# REDUCE WHAT YOU USE: INPUT-AWARE MATRIX-MULTIPLICATION PRUNING FOR LLMS

## ABSTRACT

Transformer-based language models achieve strong performance but at high computational cost, raising the question of whether their full dimensional capacity is necessary at inference. We introduce *Reduced Matrix-Multiplication* (RMM), a training-free rule that adaptively prunes feature dimensions on the fly. Given current activations, RMM scores hidden channels with simple norms, retains a controlled fraction, and performs multiplications only within this reduced subspace—yielding deterministic approximations without altering model weights. Applied uniformly across all linear operations, RMM exposes a smooth accuracy–efficiency frontier governed by a single retention ratio. Across models ranging from 1B to 70B parameters and tasks spanning question answering, reasoning, math, coding, summarization, and vision–language benchmarks, RMM achieves substantial cost reductions with minimal accuracy loss. Larger models tolerate more aggressive pruning, highlighting increasing representational redundancy at scale. These findings demonstrate that high-dimensional computations in LLMs can be systematically compressed, offering a simple and general mechanism for controllable accuracy–efficiency tradeoffs.

## 1 INTRODUCTION

Large language models (LLMs) have achieved remarkable success across a wide range of understanding and generation tasks. Typically built on the *Transformer* architecture (Vaswani et al., 2017), they process inputs token by token, incurring a substantial computational burden. In recent years, *large scaling* has become the dominant training paradigm: increasing parameters and data, widening embeddings, and deepening networks generally lead to stronger average performance (Kaplan et al., 2020). However, this trend has also made inference increasingly expensive, raising questions about whether such massive computation is always necessary for accurate predictions.

From a systems perspective, however, inference cost rises sharply with scale, and a significant share stems from high-dimensional matrix multiplications repeatedly executed across layers. In practice, computation falls into two regimes. The *prefill* stage processes the entire prompt of length $L$: projections and MLPs scale with $L$, each attention head forms an $L \times L$ score matrix, and the KV cache grows accordingly. The *decoding* stage then generates one token at a time: each step repeats the same families of matrix multiplications while interacting with all cached keys and values, leading to frequent KV reads and memory-bandwidth pressure that accumulates with context length. *These patterns highlight that a small set of operations—matrix multiplications in attention and MLPs—dominate the cost, yet it remains unclear whether all such high-dimensional computations are necessary for accurate inference.*

We therefore pose a central question: do contemporary LLMs exhibit *computational redundancy* in their high-dimensional matrix multiplications? In other words, to achieve strong performance, must every feature dimension participate in every projection, attention, and MLP operation? If not, can we exploit such redundancy at inference time to reduce cost while preserving accuracy?

Prior work has shown that transformer models contain substantial redundancy (Liu et al., 2021; Peng et al., 2023; Sajjad et al., 2023). One line of research develops *structured pruning* strategies that exploit weight- and activation-level redundancy for efficiency gains. Dependency-aware frameworks jointly remove coupled structures such as attention heads and projection channels, often with lightweight fine-tuning (Ma et al., 2023). Other methods leverage *computational invariances*, e.g.,

projecting activations onto principal components to remove low-variance directions (Ashkboos et al., 2024), or relaxing layer-wise dependencies to allow dimension selection across blocks (Gao et al., 2024). While effective, these approaches usually require retraining, rely on architecture-specific heuristics, or produce static sparsity patterns that may not adapt to inputs.

A complementary direction reduces *context redundancy* by pruning or compressing inputs and intermediate states. Some methods shorten the prefilling stage by removing redundant tokens with heuristics or learned compressors (Li et al., 2023b; Pan et al., 2024). Others operate during decoding, dynamically deleting tokens or managing the KV cache to cut memory footprint and latency (Zhang et al., 2023; Xiao et al., 2024), or leverage confidence signals to skip unnecessary generation (Fu et al., 2025). Further efforts design hardware-friendly scheduling and cache eviction policies to better utilize accelerators (Shah et al., 2024; Yuan et al., 2025). These context-level methods reduce input length or cache size, but do not directly address the high-dimensional matrix multiplications that dominate Transformer inference.

Together, prior work demonstrates that LLM redundancy can be exploited at multiple levels, but existing strategies are often retraining-intensive, architecture-specific, or confined to tokens and cache states.

In this work, we introduce *Reduced Matrix-Multiplication* RMM: a *training-free*, *input-adaptive*, and *architecture-agnostic* rule that operates directly at the level of matrix multiplications. RMM dynamically scores features for every token and every layer, selects a compact subspace on the fly. The rule applies uniformly to projections, attention, and MLP mappings, requires no parameter updates, and exposes a single budget knob that traces a smooth accuracy–efficiency frontier. Across extensive experiments, we show that RMM consistently preserves performance under substantial pruning, scales robustly from 1B to 70B parameters, and generalizes to multimodal LLMs. These results demonstrate that high-dimensional matrix multiplications in LLMs contain exploitable redundancy, and that targeting them directly offers a simple, general, and effective route to in LLM Inference.

## 2 RELATED WORK

**Empirical Evidence of Redundancy.** A growing body of research has investigated whether large transformer-based models truly require their full representational capacity, consistently revealing substantial redundancy at the level of layers, neurons, and attention mechanisms. For example, *Analyzing Redundancy* (Dalvi et al., 2020) showed that up to 85% of neurons in BERT and XLNet are redundant, and more than 90% can be pruned with little loss on downstream tasks, by combining representation-level similarity measures with neuron-level correlation clustering. *TERA* (Liu et al., 2021) found that smaller self-supervised speech encoders can match or even outperform larger ones, suggesting that added depth may not introduce new information. *LayerDrop* (Sajjad et al., 2023) demonstrated that up to 40% of layers in BERT, RoBERTa, and XLNet can be removed while preserving nearly full performance on GLUE. *HJ-Pruning* (Peng et al., 2023) extended these observations to speech models, jointly pruning convolutional and transformer components without degrading accuracy. Finally, *Redundant Transformer Stack* (Peng et al., 2023) revealed that many layers in wav2vec2 and WavLM perform nearly identical operations, and up to 45% of layers can be dropped without retraining.

Together, these studies provide strong evidence that large-scale transformer models encode substantial representational redundancy, whether measured at the level of neurons, layers, or entire blocks. While some works also propose pruning techniques to exploit this redundancy, they are often designed as offline interventions or task-specific training strategies. In contrast, our approach focuses on matrix multiplications: we directly target redundancy in high-dimensional matrix multiplications and provide a unified, input-adaptive rule that can be applied to any pre-trained model without additional training.

**Structured Model Pruning** Recent work on structured pruning goes beyond unstructured sparsification by explicitly targeting larger units such as attention heads, channels, or embedding dimensions. *LLM-Pruner* (Ma et al., 2023) introduces a dependency-aware framework that jointly prunes coupled structures (e.g., attention heads and projection channels), showing that large parameter reductions are possible with only light recovery tuning. *SliceGPT* (Ashkboos et al., 2024) instead

exploits *computational invariance* by projecting activations onto principal components and pruning low-variance directions, though at the cost of additional projection operators in residual connections. More recently, *DISP-LLM* (Gao et al., 2024) allows each block to select its own subset of embedding dimensions via binary masks produced by a lightweight hypernetwork, avoiding weight updates and revealing strong subnetworks within LLMs. Complementary to these, *Wanda* (Sun et al., 2024) identifies sparse subnetworks through one-shot pruning with calibration data, fixing a global sparse pattern without any retraining.

Together, these structured approaches demonstrate that pruning entire substructures—whether dependency-aware, projection-based, or dimension-independent—can substantially reduce the size and computation of LLMs while preserving most of their performance. However, they generally produce *static* sparsity patterns or require auxiliary training, making them less flexible at inference time. In contrast, our approach is *training-free* and *input-adaptive*, dynamically selecting subspaces at the level of matrix multiplications during inference.

**Context reduction and runtime optimization.** A complementary line of work reduces inference cost from the *context side*, either by shortening inputs before attention or by shrinking intermediate states during decoding. On the prefill path, prompt-level compression ranges from heuristics that drop low-information tokens to supervised keep/drop predictors that match prompt utility at a fraction of the length (Li et al., 2023b; Pan et al., 2024). At decoding time, heavy-hitter–plus–recency and streaming policies manage the KV cache, bounding memory and latency at long context without retraining (Zhang et al., 2023; Xiao et al., 2024). These methods reduce input length or cache size, but they do not alter the cost of the underlying high-dimensional matrix multiplications.

Context reduction has also moved *inside* the reasoning process. Confidence-aware generation prunes low-quality chains or stops early using uncertainty signals (Fu et al., 2025), while *TokenSkip* compresses chain-of-thought rationales with an auxiliary adapter. Such methods cut token usage, but typically require extra supervision or are tied to specific reasoning settings rather than being general-purpose.

Finally, orthogonal to token/KV manipulations, algorithmic and kernel-level engineering improves the efficiency of the same computation graph. Examples include exact attention kernels based on pipelining or low-precision data paths, and hardware-aligned sparse attention that translates theoretical sparsity into wall-clock gains (Shah et al., 2024; Yuan et al., 2025). These techniques rely on specialized kernels or hardware assumptions, whereas our approach remains model-agnostic.

Taken together, context- and kernel-level methods are complementary to weight-side pruning, but they target different levers of redundancy. In contrast, our work directly exploits redundancy *within the matrix multiplications themselves*, not restricted to any specific module such as attention or MLP. By targeting the atomic matrix computations that underlie virtually all operations in Transformer LMs—whether parameterized by learned weights or driven by contextual inputs—our approach applies seamlessly across both weights and contexts. This design enables a training-free, input-adaptive rule that generalizes naturally across models and tasks.

## 3 METHODOLOGY

### 3.1 RANDOMIZED LINEAR ALGEBRA FOUNDATIONS

Consider two matrices $A \in \mathbb{R}^{n \times d}$ and $B \in \mathbb{R}^{d \times m}$. Let $A^{(j)} \in \mathbb{R}^n$ denote the $j$-th column of $A$, and $B_{(j)} \in \mathbb{R}^m$ the $j$-th row of $B$. Then the product can be written as a sum of $d$ rank-one terms:

$$AB = \sum_{j=1}^{d} A^{(j)} B_{(j)}.$$

A classical randomized approximation draws $r$ indices $J_1, \ldots, J_r$ independently from $[d]$ under a probability distribution $p = (p_1, \ldots, p_d)$ with $p_j > 0$. The corresponding column–row estimator is

$$\widehat{AB} \;=\; \frac{1}{r} \sum_{t=1}^{r} \frac{1}{p_{J_t}} A^{(J_t)} B_{(J_t)} \;=\; CR,$$

with $C_{:,t} = A^{(J_t)}/\sqrt{rp_{J_t}}$ and $R_{t,:} = B_{(J_t)}/\sqrt{rp_{J_t}}$.

This estimator is *unbiased*, i.e., $\mathbb{E}[\widehat{AB}] = AB$. Its mean-square error decreases at rate $O(1/r)$. With *importance sampling*, e.g. choosing

$$p_j \ \propto \ \|A^{(j)}\|_2 \cdot \|B_{(j)}\|_2,$$

variance is minimized. More generally, when $p_j$ is chosen proportional to the *leverage scores* of $A$ and $B$, one obtains with $r = \tilde{O}(k/\varepsilon^2)$ samples a rank-$k$ approximation within error $\varepsilon$ in Frobenius or spectral norm (Drineas et al., 2006; Mahoney, 2011).

These results suggest that high-dimensional matrix multiplications can be well-approximated by selecting only a subset of columns and rows, provided the selection is aligned with their importance. Our approach builds directly on this intuition: instead of random sampling, we propose a *deterministic, input-adaptive* selection rule tailored to Transformer computations.

## 3.2 APPLICATION TO TRANSFORMER COMPUTATIONS

Building on the intuition from randomized linear algebra (Sec. 3.1), we view the high-dimensional multiplications in Transformers as structured sums of rank-one contributions. In this view, the input matrix plays the role of $A$, and its column (or row) norms provide a natural measure of importance. Our method deterministically selects a subspace of coordinates according to these norms, and then restricts the subsequent multiplication to the corresponding rows or columns of the paired matrix $B$. This yields a unified approximation rule that can be applied to *all* major matrix multiplications in the Transformer, without changing model parameters or architecture.

Formally, for a generic multiplication

$$AB = \sum_{j=1}^{d} A^{(j)} B_{(j)},$$

we define importance scores $s_j = \|A^{(j)}\|_2$, select a subset $\mathcal{S}(A) \subseteq [d]$ of the top-scoring coordinates, and restrict the computation to

$$\widetilde{AB} = \sum_{j \in \mathcal{S}(A)} A^{(j)} B_{(j)} = A P_{\mathcal{S}(A)} P_{\mathcal{S}(A)}^\top B,$$

where $P_{\mathcal{S}(A)} \in \mathbb{R}^{d \times |\mathcal{S}(A)|}$ is the projection operator that selects the columns of $A$ indexed by $\mathcal{S}(A)$ and the corresponding rows of $B$.

We quantify the degree of pruning by the *Retention Ratio (RR)*:

$$\mathrm{RR}(A) = \frac{|\mathcal{S}(A)|}{d}.$$

This ratio controls the tradeoff between accuracy and efficiency; in our experiments, we vary RR to study how different pruning levels affect performance.

**Attention.** To make the procedure concrete, we describe how the rule is applied to multi-head attention. Suppose we fix a global Retention Ratio (RR), e.g., $50\%$, so that every attention computation is pruned to half of its original dimensions. Crucially, the retained subspace is *dynamic*: at every layer, and for every input token, the important coordinates are re-computed based on the current activations. Thus the approximation changes with each sequence and step(See in figure 1).

During *prefilling*, attention scores are computed as $M = QK^\top$ with $Q, K \in \mathbb{R}^{N \times D}$, where $N$ is the token length and $D$ the embedding dimension. We treat $Q$ as the input matrix $A$, compute the norm of each column, and keep the top-$\rho_{\text{feat}} D$ dimensions according to the chosen RR $\rho_{\text{feat}}$. The same coordinates are selected in $K^\top$, and the product $QK^\top$ is then carried out on the reduced subspace.

After the score matrix $M \in \mathbb{R}^{N \times N}$ is obtained and normalized, we further restrict the value aggregation $o = \mathrm{softmax}(M)V$ by keeping only the top-$\ell$ keys per query (with ratio $\rho_{\text{tok}} = \ell/N$). In this case the selection is over rows of $M$, and the same subset is applied to $V \in \mathbb{R}^{N \times D}$, so that each query aggregates information from only a reduced set of keys.

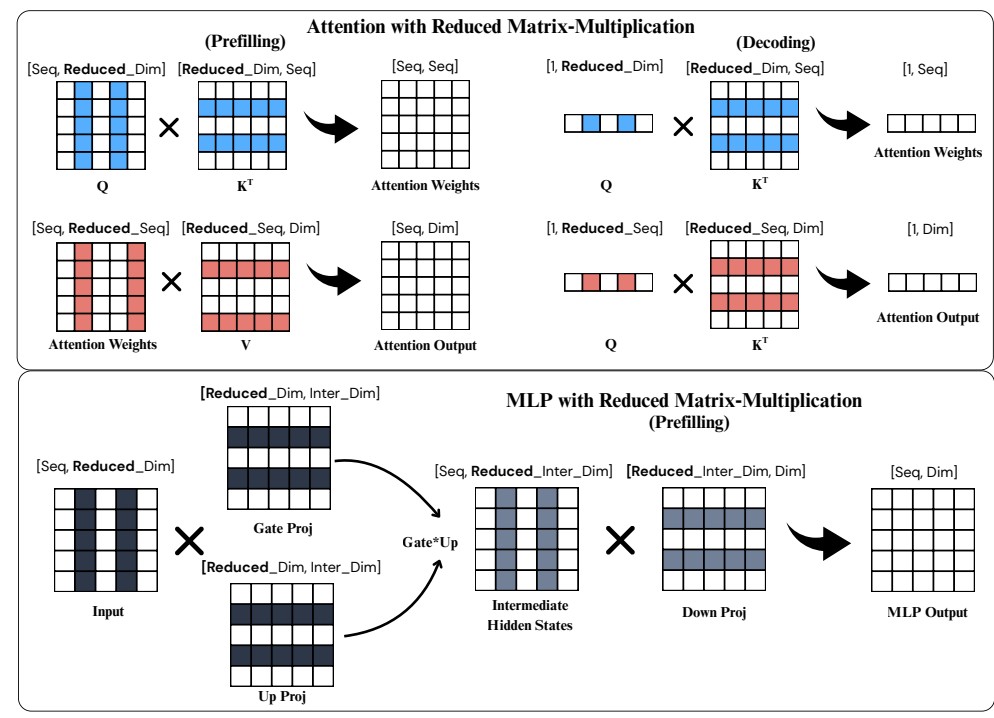

Figure 1: Application of RMM in major computations of Transformer language models.

During *decoding*, the setting is identical except that the query length reduces to $N = 1$ for the newly generated token. Here the subspace selection is again recomputed dynamically at each decoding step, leading to a smaller number of key–value lookups from the cache and hence lower memory traffic.

**MLP.** For MLP blocks, we apply the same principle to the intermediate hidden dimension. During prefilling, the input $x \in \mathbb{R}^{N \times D}$ is projected through the gate and up matrices, which expand the hidden size to $m$. We score each hidden channel by its activation norm, retain a subset $\mathcal{S}_{\text{hid}}(x)$ according to the global Retention Ratio $\rho_{\text{hid}} = |\mathcal{S}_{\text{hid}}(x)|/m$, and restrict all gate, up, and down projections to this subspace. As illustrated in Fig. 1, this amounts to computing with a reduced hidden dimension throughout the block.

### 3.3 COMPLEXITY ANALYSIS

We analyze the effect of RMM on attention and MLP computations. Let $N$ be the sequence length, $D$ the head dimension, $d_v$ the value width, and $m$ the MLP intermediate width. We denote the retention ratios as $\rho_{\text{feat}} = K/D$ (features), $\rho_{\text{tok}} = \ell/N$ (tokens), and $\rho_{\text{hid}} = r/m$ (hidden channels).

In attention, restricting score computation to $\rho_{\text{feat}}D$ features reduces the cost from $O(N^2 D)$ to $O(\rho_{\text{feat}} N^2 D)$, and limiting aggregation to $\rho_{\text{tok}} N$ keys reduces both FLOPs and KV-cache reads proportionally. In MLPs, keeping $\rho_{\text{hid}} m$ hidden channels lowers the block complexity linearly to $O(\rho_{\text{hid}} N D m)$.

Overall, RMM consistently reduces computation and memory traffic in proportion to $(\rho_{\text{feat}}, \rho_{\text{tok}}, \rho_{\text{hid}})$, offering a controllable accuracy–efficiency tradeoff. Full derivations are deferred to Appendix E.

## 4 EXPERIMENTS

### 4.1 EXPERIMENTAL SETUP

Our experiments are designed to answer four central questions:

- **RQ1: Accuracy under pruning.** Can RMM preserve task performance while reducing computation across diverse LLM benchmarks?

- **RQ2: Scaling behavior.** How does RMM behave under different pruning ratios and across model scales from 1B to 70B parameters?

- **RQ3: Comparison with baselines.** How does RMM compare to existing pruning and cache-reduction baselines on summarization and multimodal tasks?

- **RQ4: Module sensitivity.** Which components of the Transformer (QK, softmax$\times$V, MLP) can be pruned most safely, and how do different combinations affect performance?

**Models and tasks**  We evaluate RMM on a wide spectrum of pre-trained LLMs, including Llama 3.1 70B, Llama 3.1 8B, Llama 3.2 3B, Llama 3.2 1.5B (Grattafiori et al., 2024), Qwen3 32B (Yang et al., 2025), Qwen 3.1 7B, and Qwen2.5-VL-7B-Instruct (Bai et al., 2025). The benchmarks span multiple capabilities: (i) general QA and reasoning, including COPA (Gordon et al., 2012), PIQA Bisk et al. (2020), COMMONSENSEQA (Talmor et al., 2019), ARC-EASY, ARC-CHALLENGE (Clark et al., 2018), and MMLU (Hendrycks et al., 2021); (ii) text generation quality, including WIKITEXT (Merity et al., 2016), BOOKCORPUS (Zhu et al., 2015), with qualitative examples; (iii) mathematics and coding tasks, GSM8K (Cobbe et al., 2021), HUMANEVAL (Chen et al., 2021)), (iv) long-context reasoning, including RULER-CWE, RULER-HOTPOT (Hsieh et al., 2024)), (v) summarization task CNN/DAILYMAIL (Nallapati et al., 2016), (vi) and Vison Language Model tasks including: POPE (Li et al., 2023a), BLINK ART STYLE, BLINK FORENSIC DETECTION, and BLINK COUNTING (Fu et al., 2024) **All tasks are evaluated in the zero-shot setting without any task-specific fine-tuning.** For a more focused analysis of our method, the tasks reported in the main paper apply reduction only to the computation matrices within the attention modules.

**Baselines**  For summarization on CNN/DailyMail, we additionally evaluate RMM against three pruning baselines: (i) **static pruning**, where matrices are reduced according to indices determined in the prefilling stage; (ii) **random pruning**, where at each layer the matrix are selected uniformly at random; and (iii) **H2O cache reduction** (Zhang et al., 2023), which dynamically manages the KV cache during decoding.

For multimodal evaluation on Qwen2.5-VL-7B, we include static and random baselines and visualize attention maps for qualitative analysis.

**Ablations**  Since it is infeasible to test all combinations of modules, tasks, and models, we conduct controlled ablations on Llama 3.1 8B. We separately prune MLP layers, QKV projections, and attention computations and study their combinations. This isolates the effect of pruning different components and helps identify the most redundant structures. Overall, this setup ensures that RMM is assessed across *scales, tasks, baselines, and modules*, providing a comprehensive picture of its accuracy–efficiency tradeoffs.

## 4.2 MAIN RESULTS

We mainly evaluate pruning in $QK^\top$ and $\text{Softmax}(QK^\top)V$ computations, the dominant costs in attention.

**General, reasoning, math, and coding ability.**  We first evaluate models on benchmarks that measure general QA, reasoning, mathematical, and coding ability, including COPA, PIQA, COMMONSENSEQA, ARC-EASY, ARC-CHALLENGE, GSM8K, MMLU, and HUMANEVAL. For small models such as Llama-3.2-1B and Llama-3.2-3B, shows how accuracy decreases smoothly with pruning across QA tasks.Due to space limitations, we put the complete results in the appendixC. For larger models (Qwen3-1.7B, Llama-3.1-8B, Qwen3-32B, and Llama-3.1-70B), Table 1 summarizes results on the full benchmark suite. Performance degrades gracefully as the retention ratio decreases, and notably larger models tolerate more aggressive pruning while maintaining strong accuracy. For example, Llama-70B retains near-baseline performance on MMLU at 0.8 retention. This suggests that redundancy scales with model size, opening greater potential for speedup in the largest models, addressing RQ1 and RQ2.

| Model | Method | RR | Copa | ARC-C | ARC-E | PiQA | CommQA | GSM8K | MMLU | HumanEval |
|-------|--------|-----|------|-------|-------|-------|--------|-------|------|-----------|
| | Baseline | – | 72.8 | 39.8 | 69.82 | 72.69 | 47.58 | 39.87 | 55.54 | 40.24 |
| | RMM | 0.9 | 66.0 | 33.78 | 64.56 | 69.37 | 46.65 | 24.93 | 52.73 | 39.64 |
| Qwen3.1 7B | RMM | 0.8 | 64.0 | 29.77 | 57.19 | 67.19 | 47.58 | 17.58 | 47.30 | 38.41 |
| | RMM | 0.7 | 60.8 | 28.09 | 46.84 | 63.49 | 39.64 | 5.83 | 33.55 | 31.09 |
| | RMM | 0.6 | 58.4 | 25.42 | 37.02 | 59.58 | 29.89 | 2.50 | 25.97 | 20.73 |
| | RMM | 0.5 | 49.4 | 20.74 | 32.11 | 53.16 | 24.49 | 1.70 | 23.78 | 9.75 |
| | Baseline | – | 77.2 | 49.5 | 76.32 | 79.92 | 66.01 | 26.15 | 63.48 | 35.36 |
| | RMM | 0.9 | 76.6 | 48.16 | 75.26 | 79.16 | 65.44 | 24.64 | 62.20 | 35.36 |
| Llama3.1 8B | RMM | 0.8 | 77.2 | 47.49 | 75.09 | 79.05 | 64.70 | 23.73 | 60.26 | 34.75 |
| | RMM | 0.7 | 77.0 | 46.82 | 72.81 | 77.48 | 62.65 | 23.19 | 55.24 | 32.31 |
| | RMM | 0.6 | 73.4 | 37.46 | 68.60 | 77.48 | 59.38 | 14.93 | 38.62 | 26.21 |
| | RMM | 0.5 | 70.6 | 36.79 | 62.98 | 76.65 | 51.92 | 5.91 | 24.78 | 23.17 |
| | Baseline | – | 81.4 | 57.86 | 78.25 | 80.89 | 61.59 | 62.62 | 80.81 | 37.80 |
| | RMM | 0.9 | 83.6 | 55.18 | 76.14 | 80.74 | 62.24 | 62.69 | 80.01 | 40.24 |
| Qwen3 32B | RMM | 0.8 | 83.2 | 51.84 | 72.63 | 80.20 | 61.02 | 57.99 | 78.64 | 42.07 |
| | RMM | 0.7 | 82.6 | 48.49 | 69.47 | 80.58 | 60.61 | 55.11 | 77.50 | 42.68 |
| | RMM | 0.6 | 82.6 | 50.84 | 70.53 | 80.36 | 58.72 | 50.87 | 73.17 | 45.12 |
| | RMM | 0.5 | 82.2 | 46.15 | 67.19 | 77.80 | 54.87 | 39.87 | 65.09 | 46.10 |
| | Baseline | – | 84.4 | 56.19 | 78.25 | 83.24 | 57.99 | 53.67 | 75.29 | 51.21 |
| | RMM | 0.9 | 84.4 | 54.52 | 78.77 | 83.51 | 57.99 | 51.47 | 75.04 | 53.65 |
| Llama3.1 70B | RMM | 0.8 | 84.6 | 56.86 | 76.84 | 82.59 | 59.71 | 48.14 | 72.58 | 46.95 |
| | RMM | 0.7 | 81.4 | 53.18 | 74.74 | 82.48 | 59.87 | 42.75 | 67.01 | 39.63 |
| | RMM | 0.6 | 76.6 | 50.50 | 74.91 | 77.69 | 60.44 | 34.49 | 53.81 | 34.75 |
| | RMM | 0.5 | 70.2 | 41.47 | 64.21 | 73.61 | 56.84 | 19.93 | 29.73 | 18.90 |

Table 1: Performance comparison across different models and retention ratios on various benchmarks.

| RR | 1.0 | 0.9 | 0.8 | 0.7 | 0.6 | 0.5 |
|----|-----|-----|-----|-----|-----|-----|
| **WikiText Perplexity ↓** | | | | | | |
| Qwen3.1 7B | 28.64 | 31.49 | 37.80 | 54.00 | 93.52 | 219.31 |
| Llama3.1 8B | 13.39 | 14.34 | 15.22 | 17.03 | 21.35 | 32.65 |
| Qwen3 32B | 13.97 | 14.62 | 15.41 | 15.71 | 15.99 | 18.53 |
| Llama3.1 70B | 7.24 | 7.46 | 8.38 | 14.14 | 42.62 | 167.78 |
| **BookCorpus Perplexity ↓** | | | | | | |
| Qwen3.1 7B | 31.95 | 34.21 | 43.64 | 62.91 | 113.57 | 322.63 |
| Llama3.1 8B | 15.25 | 15.59 | 16.48 | 20.97 | 32.63 | 53.80 |
| Qwen3 32B | 17.43 | 17.72 | 18.12 | 18.93 | 22.21 | 29.68 |
| Llama3.1 70B | 12.20 | 12.29 | 13.06 | 19.55 | 36.55 | 126.18 |

Table 2: Perplexity across different retention ratios (RR).

**Generation quality.** We evaluate generation ability on WIKITEXT and BOOKCORPUS. As shown in Table 2, perplexity increases gradually with lower retention, with sharper growth below 0.6. Complete results are in Appendix C. Table 3 gives qualitative examples: at moderate reduction, outputs match the baseline, while at aggressive reduction they remain fluent but less detailed. Overall, RMM preserves LLM generation quality, addressing RQ1 and RQ2.

**Long-context performance.** Finally, we evaluate long-context reasoning on RULER-CWE and RULER-HOTPOT. Table 4 shows that Llama-3.1-8B maintains strong accuracy at context lengths up to 30K tokens, and pruning does not significantly harm performance.

## 4.3 COMPARISON WITH BASELINES

**Summarization tasks.** To directly address RQ3, we compare RMM against representative pruning baselines on **abstractive summarization**. Experiments are conducted on the CNN/Daily-Mail dataset, using standard metrics including ROUGE-1/2/L, ROUGE-Lsum (Ganesan, 2018), and BERTScore (Zhang et al., 2020). All methods are evaluated under the same retention level to ensure fairness, and results are summarized in Table 5.

The baselines include: (i) *Static* pruning with fixed feature selection, (ii) *Random* pruning as a lower bound, and (iii) *H2O* pruning, which retains heavy-hitter tokens plus a recent window based on portional average sequence length.

| Prompt | | Base Model | RMM Inference | |
| --- | --- | --- | --- | --- |
| | | | RR 0.8 | RR 0.6 |
| Ex.1 | The future of artificial intelligence is | here, and it's already changing the way we live and work. From self-driving cars to virtual assistants. | here, and it's already changing the way we live and work. From self-driving cars to virtual assistants. | here. It's called ChatGPT. This AI chatbot can write essays, poems, and even code. |
| Ex.2 | Tell me something about Boston. | Boston is a city in the state of Massachusetts, in the United States of America. | Boston is a city in the state of Massachusetts, in the United States of America. | I'm a native New Yorker, and I've been here for 10 years. |

Table 3: Generation comparison between base model (Llama-3.1-8B) and RMM inference under different retention ratios.

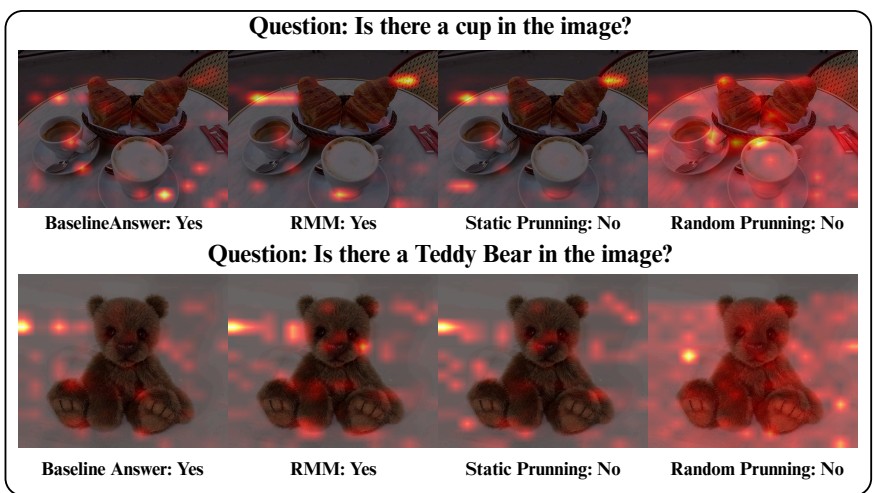

Figure 2: Attention map from the vision language model.

As shown in Table 5, our *RMM pruning* consistently outperforms all baselines across different models and pruning ratios. At a RR of 0.8, RMM matches the baseline within negligible margin while yielding substantial computational savings. Even at a more aggressive RR of 0.5, RMM outperforms static and random pruning by large margins and remains clearly stronger than H2O. These results highlight the advantage of adaptive, input-aware pruning in summarization tasks, where sequence lengths are heterogeneous and static policies struggle. Due to space limitations, we put the complete results in the appendix C.

### 4.4 EVALUATION ON VISION-LANGUAGE MODELS

**Vision-language tasks.** To address RQ3 in multimodal settings, we evaluate RMM on Qwen2.5-VL-7B using POPE, BLINK ART STYLE, BLINK FORENSIC DETECTION, and BLINK COUNTING. Table 6 shows that *Dynamic RMM* consistently outperforms static and random pruning. At 0.8 RR, performance is nearly identical to the full model, and even at 0.5 it retains high accuracy (e.g., 97.7% on Blink Forensic Detection vs. 43.3% for static and 53.0% for random).

**Attention map analysis.** Figure 2 visualizes attention for the first output token. Both the baseline and RMM focus on relevant regions (e.g., cup, teddy bear), while static pruning under-focuses and random pruning scatters attention. These qualitative patterns align with Table 6, confirming that input-adaptive pruning preserves the visual grounding needed for robust VLM performance.

### 4.5 ABLATION STUDY

**Sensitivity to different components.** To address RQ4, we study the effect of pruning different modules of Llama3.1-8B. Given the large combinatorial space across models, tasks, and targets, we focus on a representative subset, with results in Table 9 and Appendix D.

| Task | Condition | 5K | 15K | 30K |
|------|-----------|-----|------|------|
| Ruler-CWE | Baseline | 98.17 | 94.03 | 29.59 |
| | **RR 0.8** | 98.11 | 94.09 | 29.30 |
| | **RR 0.5** | 97.97 | 94.01 | 28.93 |
| Ruler-Hotpot | Baseline | 53.6 | 56.4 | 51.2 |
| | **RR 0.8** | 53.6 | 55.8 | 50.8 |
| | **RR 0.5** | 53.6 | 55.6 | 50.5 |

Table 4: Llama-3.1-8B performance on Ruler tasks across different RR and long context lengths.

| Model | Method | RR | Rouge-1 | Rouge-2 | Rouge-L | Rouge-Lsum | BERTScore |
|-------|--------|-----|---------|---------|---------|------------|-----------|
| | Baseline | – | 37.44 | 15.56 | 24.31 | 31.29 | 86.76 |
| | RMM | 0.8 | **37.54** | **15.70** | **24.23** | **31.35** | **86.72** |
| | RMM | 0.5 | **34.15** | **13.60** | **22.03** | **28.70** | **85.75** |
| | Static | 0.8 | 37.36 | 15.63 | 24.24 | 31.24 | 86.67 |
| Llama3.1 8B | Static | 0.5 | 28.01 | 9.85 | 19.28 | 24.34 | 84.02 |
| | Random | 0.8 | 6.90 | 0.60 | 6.20 | 6.69 | 77.97 |
| | Random | 0.5 | 5.67 | 0.20 | 5.23 | 5.54 | 81.42 |
| | H2O | 0.8 | 24.38 | 9.26 | 16.28 | 21.88 | 82.72 |
| | H2O | 0.5 | 24.38 | 9.26 | 16.28 | 21.88 | 82.72 |
| | Baseline | – | 36.56 | 13.05 | 22.86 | 29.75 | 85.91 |
| | RMM | 0.8 | **35.32** | **12.08** | **22.31** | **28.99** | **86.81** |
| | RMM | 0.5 | **21.91** | **5.89** | **14.54** | **18.60** | **83.33** |
| | Static | 0.8 | 34.43 | 11.63 | 22.05 | 28.47 | 86.67 |
| Qwen3-1 7B | Static | 0.5 | 4.12 | 0.05 | 3.79 | 4.01 | 78.09 |
| | Random | 0.8 | 10.23 | 0.03 | 8.04 | 9.66 | 76.31 |
| | Random | 0.5 | 6.75 | 0.06 | 6.09 | 6.54 | 76.46 |
| | H2O | 0.8 | 4.20 | 9.10 | 3.68 | 3.98 | 73.46 |
| | H2O | 0.5 | 4.20 | 9.10 | 3.68 | 3.98 | 73.46 |

Table 5: Performance comparison of pruning methods on CNN summarization.

| Task | Method | RR | Pope | Blink Art Style | Blink Forensic Detection | Blink Counting |
|------|--------|-----|------|------------------|--------------------------|----------------|
| | Baseline | – | 83.67 | 100 | 100 | 100 |
| | RMM | 0.8 | **82** | **100** | **100** | **100** |
| | Static | 0.8 | 81 | 100 | 100 | 100 |
| Qwen 2.5 7B VL | Random | 0.8 | 10.3 | 43.59 | 58.33 | 54.17 |
| | RMM | 0.5 | **67.33** | **97.44** | **97.73** | **99.17** |
| | Static | 0.5 | 63 | 91.29 | 43.33 | 79.17 |
| | Random | 0.5 | 1.3 | 41.88 | 53.03 | 43.33 |

Table 6: Performance of pruning methods on Qwen 2.5 7B VL across vision tasks

We compare pruning applied to *Q projection*, *QKV projection*, *attention*, and individual MLP projections (*up*, *gate*, *down*), as well as their combinations. Pruning only attention-side operations (e.g., QKV projection or down projection) causes modest accuracy loss, often retaining $> 90\%$ of baseline at 0.7 RR. In contrast, pruning the entire MLP block leads to sharp degradation (below $60\%$), showing MLP computations are less redundant and more critical.

These results suggest that deployments should prioritize pruning attention modules, while applying conservative strategies to MLPs, motivating hybrid pruning that combines safer targets instead of uniformly pruning all layers.

## 5 CONCLUSION

We introduced *Reduced Matrix-Multiplication* (RMM), a training-free and input-adaptive rule for pruning Transformer matrix multiplications. RMM maintains accuracy under pruning, scales robustly across model sizes, and extends naturally to multimodal models. It consistently outperforms static, random, and cache-based baselines, while ablations show that attention modules exhibit higher redundancy than MLPs. These results establish matrix multiplications as a promising lever for inference-time optimization, with future work on kernel integration and training-time extensions.

LIMITATIONS

Our study focuses on FLOPs and memory-traffic reductions, without directly measuring wall-clock latency. Realizing the practical speedups of RMM will require kernel-level integration on modern accelerators. In addition, while most of our experiments evaluate pruning on QK and attention-score×V, ablations suggest that MLP pruning is more challenging; a full exploration of MLP-side strategies is left for future work. Finally, although we evaluate RMM across a broad range of models and tasks, we cannot cover all combinations of architectures, pruning ratios, and domains, and further validation in real-world deployments is needed.

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

# Technical Appendices

## A  IMPLEMENTATION DETAILS

**Hardware and Framework.**  All experiments were run on NVIDIA GPUs (a mix of RTX A6000, RTX 6000 Ada, and L40S, each with 48GB memory). Inference was implemented in PyTorch using HuggingFace Transformers, with bf16 precision unless otherwise specified. We use the official pre-trained checkpoints without any fine-tuning or additional training. We write the attention and mlp module by ourselves and use monkey patch to replace the original module.

**Inference Setup.**  For our experiments, a global Retention Ratio (RR) was specified (e.g., from 0.9 to 0.5). RMM dynamically recomputes the retained subspace at every layer and token, applied directly within the forward pass of attention and MLP blocks. Batch sizes were chosen to fit GPU memory (typically 8–16 for 7B/8B models, 1–2 for 70B models). For long-context benchmarks such as RULER, we evaluate up to the maximum sequence length provided by the dataset (up to 30K tokens).

**Metrics.**  Perplexity is reported for language modeling tasks (WIKITEXT, BOOKCORPUS). For QA and reasoning benchmarks (COPA, PIQA, COMMONSENSEQA, MMLU, etc.), we follow the standard `lm-eval-harness` framework, running all tasks in the zero-shot setting with default accuracy as the evaluation metric. Rouge-1/2/L and BERTScore are reported for summarization (CNN/DAILYMAIL), and task-specific metrics are used for multimodal benchmarks (POPE, BLINK ART STYLE, BLINK FORENSIC DETECTION, BLINK COUNTING). All evaluations are performed in the zero-shot setting without task-specific adaptation.

## B  ALGORITHM

```python
def eager_attention_forward(Q, K, V, rho_feat, rho_tok, mask=None):
    """
    Args:
        Q, K, V : query, key, value tensors
        rho_feat : retention ratio along feature dim
        rho_tok  : retention ratio along token dim
        mask     : optional attention mask
    Returns:
        O : attention output
    """

    # === Feature-level pruning ===
    col_norm = norm(Q, axis=1)
    topk_features = topk(col_norm, k=int(rho_feat * D))
    Q, K = restrict(Q, topk_features), restrict(K, topk_features)

    # === Attention weights ===
    A = softmax(Q @ K.T / sqrt(d))
    if mask is not None:
        A += mask

    # === Token-level pruning ===
    row_norm = norm(A, axis=1)
    topk_tokens = topk(row_norm, k=int(rho_tok * Lk))
    A, V = restrict(A, topk_tokens), restrict(V, topk_tokens)

    # === Output ===
    O = A @ V
    return O
```

Listing 1: Attention forward with dual RMM for short seq.

```python
def eager_attention_forward(Q, K, V, rho_feat, rho_tok, mask=None,
                            k_tile=1024, q_tile=None, dropout=0.0):
    """
    Three-pass streaming implementation:
      1) Feature Top-k: select rho_feat * D dimensions per head
      2) Token Top-k: select rho_tok * Lk keys based on softmax column
         norms
      3) Final output: restrict to selected keys without re-normalization
    """

    # --- Expand KV groups to per-head ---
    K = repeat_kv(K)    # [B,H,Lk,D]
    V = repeat_kv(V)    # [B,H,Lk,Dv]

    # --- Feature Top-k ---
    col_norm = norm(Q, axis=2)
    feat_idx = topk(col_norm, k=int(rho_feat * D))
    Q, K = restrict(Q, feat_idx), restrict(K, feat_idx)

    # --- PASS-1: softmax denominator (streaming, stable) ---
    for each query block Q_blk in Q:
        for each key block K_blk in K:
            logits = Q_blk @ K_blk.T / sqrt(d)
            if mask: logits += mask
            update row-wise max m and normalizer l

    # --- PASS-2: accumulate column scores for Token Top-k ---
    col_score = zeros(B,H,Lk)
    for each (Q_blk, K_blk) block:
        logits = Q_blk @ K_blk.T / sqrt(d)
        if mask: logits += mask
        W = exp(logits - m) / l
        col_score += sum(W**2, axis=query)

    token_idx = topk(col_score, k=int(rho_tok * Lk))

    # --- PASS-3: final output restricted to selected keys ---
    out = zeros(B,H,Lq,Dv)
    for each (Q_blk, K_blk, V_blk) block:
        logits = Q_blk @ K_blk.T / sqrt(d)
        if mask: logits += mask
        W = exp(logits - m) / l
        W = mask_columns(W, token_idx)
        out += W @ V_blk

    return out.transpose(1,2)   # [B,Lq,H,Dv]
```

Listing 2: Streaming eager attention with dual Top-k pruning (long-sequence version).

```python
class DynLinearCols(nn.Module):
    def __init__(self, in_f, out_f, k_ratio=0.5, bias=True):
        super().__init__()
        self.weight = nn.Parameter(torch.empty(out_f, in_f))
        self.bias   = nn.Parameter(torch.empty(out_f)) if bias else None
        nn.init.xavier_uniform_(self.weight)
        if self.bias is not None:
            nn.init.uniform_(self.bias, -1/in_f**0.5, 1/in_f**0.5)
        self.k_ratio = k_ratio

    def forward(self, x):
        B, S, D = x.shape
        K = max(1, int(D * self.k_ratio))
        col_norm = x.norm(dim=1)
        _, idx = torch.topk(col_norm, K, dim=-1)
```

```
16          outputs = []
17          for b in range(B):
18              x_b = x[b, :, idx[b]]
19              w_b = self.weight[:, idx[b]]
20              y_b = F.linear(x_b, w_b, self.bias)
21              outputs.append(y_b)
22          return torch.stack(outputs, dim=0)
23
24
25  class DynLlamaMLP(nn.Module):
26      def __init__(self, cfg):
27          super().__init__()
28          hs, isz = cfg.hidden_size, cfg.intermediate_size
29          self.gate_proj = DynLinearCols(hs, isz, bias=cfg.mlp_bias)
30          self.up_proj   = DynLinearCols(hs, isz, bias=cfg.mlp_bias)
31          self.down_proj = nn.Linear(isz, hs, bias=cfg.mlp_bias)
32          self.act_fn    = ACT2FN[cfg.hidden_act]
33
34      def forward(self, x):
35          g = self.act_fn(self.gate_proj(x))
36          u = self.up_proj(x)
37          return self.down_proj(g * u)
38
39
40  def replace_llama_mlp(module, cfg):
41      from transformers.models.llama.modeling_llama import LlamaMLP as
              HFMLP
42      for name, child in list(module.named_children()):
43          if isinstance(child, HFMLP):
44              new = DynLlamaMLP(cfg)
45              new = new.to(child.gate_proj.weight.device, dtype=child.
                  gate_proj.weight.dtype)
46              with torch.no_grad():
47                  for old_l, new_l in zip(
48                      [child.gate_proj, child.up_proj, child.down_proj],
49                      [new.gate_proj , new.up_proj , new.down_proj ]):
50                      new_l.weight.copy_(old_l.weight)
51                      if old_l.bias is not None:
52                          new_l.bias.copy_(old_l.bias)
53              setattr(module, name, new)
54          else:
55              replace_llama_mlp(child, cfg)
```

Listing 3: RMM For MLP.

```
1  class DynLinearCols(nn.Module):
2      def __init__(self, in_f, out_f, k_ratio=0.33, bias=True):
3          super().__init__()
4          self.weight = nn.Parameter(torch.empty(out_f, in_f))
5          self.bias   = nn.Parameter(torch.empty(out_f)) if bias else None
6          nn.init.xavier_uniform_(self.weight)
7          if self.bias is not None:
8              nn.init.uniform_(self.bias, -1/in_f**0.5, 1/in_f**0.5)
9          self.k_ratio = k_ratio
10
11     def forward(self, x):  # x: [B, S, D]
12          B, S, D = x.shape
13          k_ratio = GLOBAL_K_RATIO if GLOBAL_K_RATIO is not None else self.
              k_ratio
14          K = max(1, int(D * k_ratio))
15          with torch.no_grad():
16              col_norm = x.norm(dim=1)                    # [B, D]
17              _, idx = torch.topk(col_norm, K, dim=-1)  # [B, K]
18          outputs = []
19          for b in range(B):
```

```
20              x_b = x[b, :, idx[b]]                     # [S, K]
21              w_b = self.weight[:, idx[b]]              # [O, K]
22              y_b = F.linear(x_b, w_b, self.bias)       # [S, O]
23              outputs.append(y_b)
24          return torch.stack(outputs, dim=0)            # [B, S, O]
25
26
27  def replace_attention_projections(module):
28      from transformers.models.llama.modeling_llama import LlamaAttention
            as HFAttention
29      for name, child in list(module.named_children()):
30          if isinstance(child, HFAttention):
31              old_q, old_k, old_v = child.q_proj, child.k_proj, child.
                    v_proj
32              new_q = DynLinearCols(old_q.in_features, old_q.out_features,
                    bias=old_q.bias is not None)
33              new_k = DynLinearCols(old_k.in_features, old_k.out_features,
                    bias=old_k.bias is not None)
34              new_v = DynLinearCols(old_v.in_features, old_v.out_features,
                    bias=old_v.bias is not None)
35              for old_l, new_l in zip([old_q, old_k, old_v], [new_q, new_k,
                     new_v]):
36                  new_l = new_l.to(old_l.weight.device, dtype=old_l.weight.
                        dtype)
37                  with torch.no_grad():
38                      new_l.weight.copy_(old_l.weight)
39                      if old_l.bias is not None:
40                          new_l.bias.copy_(old_l.bias)
41              child.q_proj = new_q
42              child.k_proj = new_k
43              child.v_proj = new_v
44          else:
45              replace_attention_projections(child)
```

Listing 4: RMM For QKV Projection.

## C  SUPPLEMENTARY RESULTS

### C.1  ADDITIONAL RESULTS ON QA AND LANGUAGE MODELING

To complement the QA and generation analysis in Section 4.2, we provide full perplexity results on
WIKITEXT and BOOKCORPUS for smaller-scale models. In addition to the 7B/8B/32B/70B models reported in the main paper, we include Llama-3.2-1B and Llama-3.2-3B. Figure 3 and Table 7
present the complete results under different retention ratios (RR).

The results show the same trend as larger models: perplexity increases gradually as RR decreases,
with a sharp degradation once RR drops below 0.6. Moreover, the 3B model consistently shows
greater robustness than the 1B model (Figure 3), reinforcing our claim that representational redundancy grows with scale.

### C.2  ADDITIONAL RESULTS ON SUMMARIZATION

We also expand the summarization results on CNN/DAILYMAIL beyond those in the main paper.
Table 8 includes Llama-3.2-1B and 3B alongside the larger 7B/8B models.

The trends mirror those observed in Section 4.3: RMM matches the baseline at RR = 0.8 across all
scales, while static and random pruning degrade severely. At RR = 0.5, smaller models drop more
sharply, but RMM remains consistently better than all baselines. This confirms that our method
preserves summarization quality even in small-scale models, while redundancy increases with size,
enabling more aggressive pruning at larger scales.

| RR | 1.0 | 0.9 | 0.8 | 0.7 | 0.6 | 0.5 |
|---|---|---|---|---|---|---|
| **WikiText Perplexity ↓** | | | | | | |
| Llama3.2 1B | 20.04 | 20.66 | 22.77 | 31.29 | 68.52 | 151.64 |
| Llama3.2 3B | 15.89 | 16.23 | 17.01 | 18.82 | 25.04 | 42.54 |
| Qwen3.1 7B | 28.64 | 31.49 | 37.80 | 54.00 | 93.52 | 219.31 |
| Llama3.1 8B | 13.39 | 14.34 | 15.22 | 17.03 | 21.35 | 32.65 |
| Qwen3 32B | 13.97 | 14.62 | 15.41 | 15.71 | 15.99 | 18.53 |
| Llama3.1 70B | 7.24 | 7.46 | 8.38 | 14.14 | 42.62 | 167.78 |
| **BookCorpus Perplexity ↓** | | | | | | |
| Llama3.1 1B | 21.18 | 22.03 | 24.39 | 39.63 | 96.81 | 192.81 |
| Llama3.2 3B | 17.79 | 18.07 | 18.86 | 21.84 | 34.47 | 54.53 |
| Qwen3.1 7B | 31.95 | 34.21 | 43.64 | 62.91 | 113.57 | 322.63 |
| Llama3.1 8B | 15.25 | 15.59 | 16.48 | 20.97 | 32.63 | 53.80 |
| Qwen3 32B | 17.43 | 17.72 | 18.12 | 18.93 | 22.21 | 29.68 |
| Llama3.1 70B | 12.20 | 12.29 | 13.06 | 19.55 | 36.55 | 126.18 |

Table 7: Perplexity matrix across RR

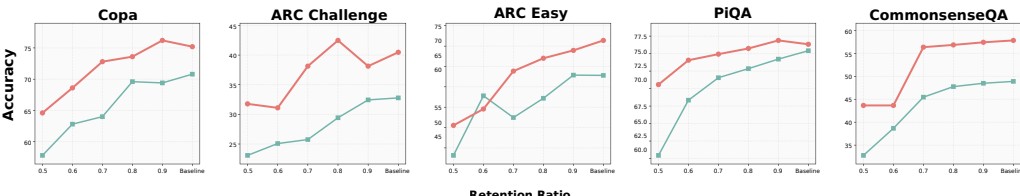

Figure 3: Llama-3.2-3B and Llama-3.2-1B of Different Tasks and RR. Red Line is Llama-3.2-3B

## D    ABLATION STUDY DETAILS

**Sensitivity to different components.**    To further investigate RQ4, we provide comprehensive ablation results on Llama3.1-8B, summarized in Table **??**. This analysis examines pruning applied to individual projections ($Q$, $K$, $V$), combinations of QKV and attention computations, and individual MLP submodules (*up*, *gate*, *down*), as well as pruning the entire MLP block or combinations with attention. By presenting the full set of results, we highlight consistent patterns across different targets and retention ratios.

**Attention-side pruning.**    Pruning restricted to attention-related operations (Q-only, QKV projections, or the attention matrix) generally yields moderate accuracy degradation. Performance remains above 90% of the baseline even at RR = 0.7 for most QA tasks, confirming that attention modules contain considerable redundancy. Among these, pruning Q or QKV projections is more stable, while directly pruning the attention matrix leads to steeper drops at lower retention ratios. These results support our claim that attention is a safe target for aggressive pruning.

**MLP-side pruning.**    In contrast, pruning within MLPs has a more severe effect. Removing hidden channels in the *up*, *gate*, or *down* projections degrades performance faster than in attention. The *down* projection is slightly more robust, but once the entire MLP block is pruned, accuracy collapses rapidly: at RR = 0.5, performance drops to nearly 40% of the baseline average. This suggests that MLP computations are less redundant and contribute critically to model accuracy.

**Hybrid strategies.**    When attention and MLP pruning are combined, degradation compounds quickly, with accuracy falling below 50% at RR = 0.6. This indicates that uniformly applying high pruning ratios across all modules is not effective. Instead, hybrid strategies should prioritize pruning attention-side computations more aggressively, while applying more conservative pruning to MLPs.

| Model | Method | RR | Rouge-1 | Rouge-2 | Rouge-L | Rouge-Lsum | BERTScore |
|---|---|---|---|---|---|---|---|
| Llama3.1 8B | Baseline | – | 37.44 | 15.56 | 24.31 | 31.29 | 86.76 |
| | RMM | 0.8 | **37.54** | **15.70** | **24.23** | **31.35** | **86.72** |
| | RMM | 0.5 | **34.15** | **13.60** | **22.03** | **28.70** | **85.75** |
| | Static | 0.8 | 37.36 | 15.63 | 24.24 | 31.24 | 86.67 |
| | Static | 0.5 | 28.01 | 9.85 | 19.28 | 24.34 | 84.02 |
| | Random | 0.8 | 6.90 | 0.60 | 6.20 | 6.69 | 77.97 |
| | Random | 0.5 | 5.67 | 0.20 | 5.23 | 5.54 | 81.42 |
| | H2O | 0.8 | 24.38 | 9.26 | 16.28 | 21.88 | 82.72 |
| | H2O | 0.5 | 24.38 | 9.26 | 16.28 | 21.88 | 82.72 |
| Llama3.2 1B | Baseline | – | 36.51 | 15.24 | 23.55 | 30.59 | 86.41 |
| | RMM | 0.8 | **37.27** | **15.71** | **23.62** | **30.96** | **86.35** |
| | RMM | 0.5 | **11.90** | **2.35** | **9.80** | **11.15** | **79.55** |
| | Static | 0.8 | 36.77 | 15.36 | 23.39 | 30.68 | 86.35 |
| | Static | 0.5 | 4.73 | 0.07 | 4.26 | 4.58 | 75.53 |
| | Random | 0.8 | 7.30 | 0.02 | 6.58 | 7.08 | 79.15 |
| | Random | 0.5 | 6.26 | 0.40 | 5.53 | 6.01 | 76.85 |
| | H2O | 0.8 | 23.22 | 8.12 | 15.56 | 20.30 | 83.76 |
| | H2O | 0.5 | 23.22 | 8.12 | 15.56 | 20.30 | 83.76 |
| Llama3.2 3B | Baseline | – | 36.72 | 15.08 | 23.56 | 30.63 | 86.55 |
| | RMM | 0.8 | **36.72** | **15.16** | **23.58** | **30.66** | **86.54** |
| | RMM | 0.5 | **28.31** | **9.83** | **19.45** | **24.60** | **84.74** |
| | Static | 0.8 | 35.96 | 14.81 | 23.24 | 30.13 | 86.31 |
| | Static | 0.5 | 19.91 | 6.06 | 14.69 | 17.83 | 82.06 |
| | Random | 0.8 | 6.67 | 0.39 | 5.93 | 6.44 | 79.72 |
| | Random | 0.5 | 5.85 | 0.04 | 5.33 | 5.70 | 77.09 |
| | H2O | 0.8 | 19.49 | 6.32 | 13.83 | 17.03 | 82.78 |
| | H2O | 0.5 | 19.49 | 6.32 | 13.83 | 17.03 | 82.78 |
| Qwen3-1 7B | Baseline | – | 36.56 | 13.05 | 22.86 | 29.75 | 85.91 |
| | RMM | 0.8 | **35.32** | **12.08** | **22.31** | **28.99** | **86.81** |
| | RMM | 0.5 | **21.91** | **5.89** | **14.54** | **18.60** | **83.33** |
| | Static | 0.8 | 34.43 | 11.63 | 22.05 | 28.47 | 86.67 |
| | Static | 0.5 | 4.12 | 0.05 | 3.79 | 4.01 | 78.09 |
| | Random | 0.8 | 10.23 | 0.03 | 8.04 | 9.66 | 76.31 |
| | Random | 0.5 | 6.75 | 0.06 | 6.09 | 6.54 | 76.46 |
| | H2O | 0.8 | 4.20 | 9.10 | 3.68 | 3.98 | 73.46 |
| | H2O | 0.5 | 4.20 | 9.10 | 3.68 | 3.98 | 73.46 |

Table 8: Performance comparison of different pruning methods on CNN summarization task. RMM consistently outperforms baseline methods across different models and retention ratios.

**Takeaways.** Overall, the ablations provide three key insights: (i) Attention modules exhibit high redundancy and can sustain substantial pruning with minimal loss. (ii) MLP blocks, while larger in parameter count, are more sensitive to pruning, particularly when entire blocks are reduced. (iii) Effective deployment should adopt heterogeneous pruning policies that allocate higher budgets to attention and lower budgets to MLPs. These findings reinforce the general message of our method: redundancy exists across multiple components, but its distribution is uneven, and exploiting this structure is critical for robust efficiency gains.

# E    EFFICIENCY ANALYSIS

## E.1    COMPLEXITY ANALYSIS

We provide a detailed analysis of how RMM affects the two dominant regimes of Transformer inference: (i) attention and (ii) MLP computation. Notation: let $N$ be the sequence length, $D$ the head dimension, $d_v$ the value dimension, and $m$ the intermediate hidden width of the MLP. For retention ratios we write $\rho_{\text{feat}} = K/D$ for features, $\rho_{\text{tok}} = \ell/N$ for tokens, and $\rho_{\text{hid}} = r/m$ for hidden channels.

**(i) Attention.**    Self-attention consists of two main multiplications: the score matrix $QK^\top$ and the value aggregation $\text{softmax}(QK^\top)V$. Computing $QK^\top$ over $N$ tokens scales as $O(N^2 D)$. Restricting to $\rho_{\text{feat}} D$ features reduces this to $O(\rho_{\text{feat}} N^2 D)$, with both FLOPs and activation traffic reduced proportionally to $\rho_{\text{feat}}$. For value aggregation, the cost drops from $O(N^2 d_v)$ to $O(N\ell d_v) = O(\rho_{\text{tok}} N^2 d_v)$ by keeping only $\rho_{\text{tok}} N$ keys per query. Thus, attention complexity scales down multiplicatively with $(\rho_{\text{feat}}, \rho_{\text{tok}})$.

| Method | Ratio | Copa | ARC-C | ARC-E | PiQA | CommQA | AVG |
|---|---|---|---|---|---|---|---|
| **Prune Q Projection** | Baseline | 77.2 | 49.5 | 76.32 | 79.92 | 66.01 | 69.79 |
| | 0.9 | 77.4 | 48.16 | 76.67 | 79.6 | 66.18 | 69.60 |
| | 0.8 | 77.6 | 49.83 | 76.84 | 79.71 | 66.42 | 70.08 |
| | 0.7 | 77.6 | 50.84 | 76.67 | 79.82 | 65.11 | 70.01 |
| | 0.6 | 77.2 | 47.83 | 76.14 | 78.94 | 65.68 | 69.16 |
| | 0.5 | 77.2 | 44.15 | 73.86 | 79.27 | 64.54 | 67.80 |
| **Prune QKV Projection** | Baseline | 77.2 | 49.5 | 76.32 | 79.92 | 66.01 | 69.79 |
| | 0.9 | 76.6 | 48.16 | 75.26 | 79.16 | 65.44 | 68.92 |
| | 0.8 | 77.2 | 47.49 | 75.09 | 79.05 | 64.70 | 68.71 |
| | 0.7 | 77.0 | 46.82 | 72.81 | 77.48 | 62.65 | 67.35 |
| | 0.6 | 73.4 | 37.46 | 68.60 | 77.48 | 59.38 | 63.26 |
| | 0.5 | 70.6 | 36.79 | 62.98 | 76.65 | 51.92 | 59.79 |
| **Prune Attention** | Baseline | 77.2 | 49.5 | 76.32 | 79.92 | 66.01 | 69.79 |
| | 0.9 | 78.2 | 48.49 | 75.79 | 79.22 | 65.57 | 69.45 |
| | 0.8 | 74.8 | 47.83 | 75.44 | 79.05 | 64.95 | 68.41 |
| | 0.7 | 78.0 | 44.15 | 70.70 | 78.89 | 63.14 | 66.98 |
| | 0.6 | 76.6 | 39.13 | 68.42 | 78.73 | 59.71 | 64.52 |
| | 0.5 | 74.6 | 32.11 | 62.11 | 74.76 | 54.22 | 59.56 |
| **Prune Attention&Q** | Baseline | 77.2 | 49.5 | 76.32 | 79.92 | 66.01 | 69.79 |
| | 0.9 | 77.4 | 48.16 | 76.67 | 79.60 | 66.18 | 69.60 |
| | 0.8 | 77.6 | 49.83 | 76.84 | 79.71 | 66.42 | 70.08 |
| | 0.7 | 77.6 | 50.84 | 76.67 | 79.82 | 65.11 | 70.01 |
| | 0.6 | 77.2 | 47.83 | 76.14 | 78.94 | 65.68 | 69.16 |
| | 0.5 | 77.2 | 44.15 | 73.86 | 79.27 | 64.54 | 67.80 |
| **Prune Attention&QKV** | Baseline | 77.2 | 49.5 | 76.32 | 79.92 | 66.01 | 69.79 |
| | 0.9 | 76.6 | 48.20 | 75.30 | 79.20 | 65.40 | 68.94 |
| | 0.8 | 77.2 | 47.50 | 75.10 | 79.10 | 64.70 | 68.72 |
| | 0.7 | 77.0 | 46.80 | 72.80 | 78.10 | 62.70 | 67.48 |
| | 0.6 | 73.4 | 37.50 | 68.60 | 77.50 | 59.40 | 63.28 |
| | 0.5 | 70.6 | 36.80 | 63.00 | 76.60 | 51.90 | 59.78 |
| **Prune MLP Up** | Baseline | 77.2 | 49.5 | 76.32 | 79.92 | 66.01 | 69.79 |
| | 0.9 | 73.0 | 46.49 | 70.70 | 78.07 | 60.20 | 65.69 |
| | 0.8 | 73.6 | 38.46 | 62.98 | 76.17 | 58.31 | 61.90 |
| | 0.7 | 68.2 | 41.47 | 60.00 | 74.59 | 55.12 | 59.88 |
| | 0.6 | 69.0 | 34.55 | 58.07 | 73.72 | 53.15 | 57.70 |
| | 0.5 | 66.8 | 28.76 | 51.05 | 69.15 | 46.44 | 52.44 |
| **Prune MLP Gate** | Baseline | 77.2 | 49.5 | 76.32 | 79.92 | 66.01 | 69.79 |
| | 0.9 | 73.6 | 44.82 | 74.74 | 79.33 | 60.69 | 66.64 |
| | 0.8 | 74.2 | 47.16 | 70.00 | 77.42 | 60.03 | 65.76 |
| | 0.7 | 74.8 | 41.81 | 69.12 | 75.35 | 59.30 | 64.08 |
| | 0.6 | 74.4 | 42.81 | 66.49 | 74.27 | 56.51 | 62.90 |
| | 0.5 | 69.4 | 34.11 | 63.86 | 71.49 | 50.61 | 57.89 |
| **Prune MLP Down** | Baseline | 77.2 | 49.5 | 76.32 | 79.92 | 66.01 | 69.79 |
| | 0.9 | 76.8 | 47.16 | 73.33 | 78.84 | 61.02 | 67.43 |
| | 0.8 | 76.0 | 46.49 | 73.51 | 78.89 | 59.71 | 66.92 |
| | 0.7 | 74.8 | 45.48 | 72.81 | 77.20 | 58.48 | 65.75 |
| | 0.6 | 74.0 | 42.81 | 66.84 | 76.71 | 53.71 | 62.81 |
| | 0.5 | 72.0 | 40.80 | 64.04 | 76.22 | 53.73 | 61.36 |
| **Prune Whole MLP** | Baseline | 77.2 | 49.5 | 76.32 | 79.92 | 66.01 | 69.79 |
| | 0.9 | 69.6 | 43.81 | 67.02 | 76.66 | 58.23 | 63.06 |
| | 0.8 | 68.6 | 37.46 | 61.75 | 75.39 | 54.55 | 59.55 |
| | 0.7 | 70.0 | 31.77 | 57.54 | 71.44 | 48.89 | 55.93 |
| | 0.6 | 65.2 | 25.08 | 48.95 | 63.76 | 42.42 | 49.08 |
| | 0.5 | 53.6 | 23.75 | 35.44 | 57.56 | 31.04 | 40.28 |
| **Prune MLP&Attention** | Baseline | 77.2 | 49.5 | 76.32 | 79.92 | 66.01 | 69.79 |
| | 0.9 | 69.6 | 41.47 | 65.61 | 77.15 | 58.97 | 62.56 |
| | 0.8 | 69.8 | 35.12 | 56.32 | 73.72 | 55.86 | 58.16 |
| | 0.7 | 68.4 | 33.11 | 52.28 | 67.85 | 48.73 | 54.07 |
| | 0.6 | 56.6 | 27.76 | 44.56 | 57.51 | 36.69 | 44.62 |
| | 0.5 | 56.4 | 25.08 | 31.05 | 52.88 | 28.01 | 38.68 |

Table 9: Comprehensive comparison of different pruning methods on Llama-8B across multiple benchmarks. All methods show similar patterns with gradual performance degradation as the retention ratio decreases.

During autoregressive decoding at step $t$, each new token attends to $t$ cached keys and values. The score computation reduces from $O(tD)$ to $O(\rho_{\text{feat}}tD)$. Likewise, reading $V$ vectors reduces from $O(td_v)$ to $O(\rho_{\text{tok}}td_v)$. This lowers both FLOPs and memory traffic, while KV-cache reads shrink by $\rho_{\text{tok}}$—a critical factor since cache bandwidth often dominates latency in long-context decoding.

**(ii) MLPs.** A standard two-layer feed-forward network with hidden width $m$ has cost $O(NDm + NmD)$. Applying RMM to retain only $\rho_{\text{hid}}m$ hidden channels reduces this to $O(\rho_{\text{hid}}NDm)$, saving computation linearly with $\rho_{\text{hid}}$. Importantly, since the same channel subset is used in the *gate*, *up*, and *down* projections, the pruning applies consistently across all multiplications without introducing auxiliary parameters or kernels. This yields reductions in both compute and memory proportional to $\rho_{\text{hid}}$.

**Comparison to other pruning strategies.** Unlike token-level pruning (which reduces sequence length) or weight pruning (which reduces parameters), RMM targets the *matrix multiplications themselves*. This is the atomic operation underlying both attention and MLP blocks, and thus improvements directly translate to reductions in FLOPs, memory movement, and cache bandwidth. Since FLOP count, memory traffic, and cache I/O all scale with $(\rho_{\text{feat}}, \rho_{\text{tok}}, \rho_{\text{hid}})$, RMM exposes a smooth tradeoff between accuracy and efficiency that can be tuned by a single retention ratio.

**Practical implications.** In our implementation, reductions are realized at the level of activation selection. If integrated with custom CUDA kernels that skip the pruned dimensions (rather than simply masking after multiplication), the theoretical savings above can directly yield wall-clock speedups and memory reductions. This means that RMM, once implemented natively, can achieve real acceleration comparable to other efficiency methods such as FlashAttention or block-sparse kernels.

**Summary.** Across both attention and MLPs, RMM consistently reduces FLOPs, activation traffic, and cache bandwidth in direct proportion to the retention ratios. Because the method operates deterministically and input-adaptively, it provides fine-grained control of the accuracy–efficiency frontier and is fully compatible with future kernel-level implementations that can convert the savings into end-to-end inference speedups.

