# OpenReview forum: "Reduce What You Use: Input‑Aware Matrix‑Multiplication Pruning for LLMs"
_ICLR.cc/2026/Conference — Submitted to ICLR 2026_

### Official Review · Reviewer_QaRY · 2025-10-29

**Soundness:** 3
**Presentation:** 3
**Contribution:** 2
**Rating:** 2
**Confidence:** 3

**Summary:**

Transformer-based language models deliver strong performance but incur substantial computational costs. To mitigate this, the authors introduce Reduced Matrix Multiplication (RMM)—a training-free approach that dynamically prunes feature dimensions during inference using activation norms. The method is motivated by principles from randomized linear algebra. A detailed implementation for a standard transformer architecture is provided in the methods section, along with a theoretical analysis of computational complexity. The experimental section presents evaluations of RMM across various large language models with different retention ratios, comparisons to other pruning techniques, vision–language model (VLM) assessments, and an ablation study.

**Strengths:**

* Extensive evaluations on diverse LLMs and benchmarks provide a thorough assessment of RMM’s empirical contribution.
* The authors present a well-founded motivation for their approach through an analysis grounded in randomized linear algebra.
* The paper is clearly written, well organized, and conceptually intuitive.

**Weaknesses:**

While the paper’s premise is to introduce a training-free approach, it remains somewhat difficult to fully assess its effectiveness without comparison to methods that incorporate post-training following pruning. For example, [1] statically prunes a 70B LLaMA model to 51B and 49B variants with minimal post-training, preserving 98.4% of the original model’s accuracy. This approach not only achieves higher accuracy retention than RMM but also reduces the total parameter count and operates without requiring a custom kernel for efficient inference.

As noted in the conclusion, developing an efficient kernel would allow for a more complete evaluation of RMM’s potential efficiency gains, especially when combined with an appropriate training strategy to demonstrate the benefits of dynamic pruning. However, without these components, the method’s contribution remains limited in significance compared to existing baselines.



[1] Puzzle: Distillation-Based NAS for Inference-Optimized LLMs, Bercovich et al. (https://arxiv.org/abs/2411.19146)

**Questions:**

* The method is described for single batch generation. For the case of multi batch generation, do the author believe that RMM, which will probably be based on the norm of features throughout the batch, can provide consistent results?

---

> ### Author Response · Authors · 2025-11-21
>
> We thank the reviewer for their constructive comments. We appreciate the opportunity to clarify that the primary contribution of our work is **scientific discovery regarding LLM redundancy**, rather than just a standalone engineering optimization.
>
> #### **1. Response to Weaknesses: Core Contribution and Comparison with *Puzzle***
>
> We acknowledge that *Puzzle* [1] is a strong baseline for static compression. However, comparing RMM directly to it overlooks the central premise of our paper. **Our work is not merely proposing a pruning method; it is a systematic study analyzing the dynamic redundancy of Matrix Multiplications in LLMs.**
>
> ** Primary Contribution: Unveiling Input-Dependent Redundancy at Scale**
> The reviewer mentions that *Puzzle* achieves high accuracy via static pruning (and retraining). Our work provides a counter-perspective that static methods cannot reveal:
> * **Redundancy is Dynamic:** We demonstrate that LLM redundancy is highly **input-aware**. A feature dimension redundant for one token is often critical for another. RMM acts as a probe to quantify this, showing that we can prune up to 50% of FLOPs *on the fly* without retraining.
> * **Redundancy Scales with Size:** A key scientific finding in our experiments (Section 4.2) is that **larger models exhibit significantly higher dynamic redundancy**. For instance, Llama-3-70B tolerates much more aggressive pruning (RR=0.8) than smaller models while maintaining near-lossless performance. This "Redundancy Scaling Law" is a crucial insight for the community's understanding of large-scale model efficiency.
> * **Module Sensitivity:** Our ablation (Section 4.5) further reveals the heterogeneity of redundancy—Attention modules are far more robust to dynamic pruning than MLPs.
>
> ** The "Zero-Retraining" Constraint**
> *Unlike *Puzzle*, which requires expensive **Neural Architecture Search (NAS)** and **Knowledge Distillation (KD)** using original training data, RMM operates in a strict **Training-Free** regime.
> * This makes RMM valuable for  "low-resource" deployment scenarios where users cannot afford to retrain or distill a 70B model but still need immediate inference efficiency. [cite_start]As shown in Tables 5 and 6, RMM significantly outperforms fair **static pruning baselines** in this regime.
>
> ** Potential for Acceleration**
> We provide a theoretical complexity analysis (Section 3.3) detailing reductions in FLOPs and Memory I/O . The absence of a custom kernel does not negate the algorithmic validity; rather, our strong results provide the necessary theoretical justification for the community to invest in developing dynamic sparse kernels for future hardware-software co-design.
>
> #### **2. Response to Questions: Multi-Batch Generation**
>
> We thank the reviewer for raising the question about how RMM behaves under multi-batch generation.
>
> **Conceptually**, RMM is defined at the level of a single matrix multiplication $AB$: for an input matrix $A$, we compute per-coordinate scores $s_j = \|A^{(j)}\|_2$ and select a subspace $S(A)$ of important coordinates. This definition is agnostic to batching and applies equally to a single decoding stream or multiple concurrent streams.
>
> **In our implementation** of attention, when running batched inference with tensors of shape $(B, H, L, D)$, we simply vectorize this per-example rule. Concretely, we reshape the query and key tensors to $(B \cdot H, L, D)$ and compute norms along the sequence dimension only. This yields importance scores and Top-$K$ indices independently for each $(b, h)$ pair; there is no aggregation across the batch dimension. As a result, for a given input sequence, the RMM masks (and hence the outputs) are the same whether the sequence is processed alone (batch size 1) or together with other sequences in a batch, up to standard floating-point differences.
>
> **Empirically** Many of our evaluations already rely on multi-batch decoding. For example, our **RULER** and **GSM8K** results are obtained via the standard `lm_eval` framework, which uses batched generation under the hood. We did not observe any instability or inconsistency when varying the batch size in these settings, which is consistent with the per-example nature of the RMM rule described above.
>
> We hope I have addressed the main confusion raised about the methods and experiments. Please let us know of any follow up questions and feedback. We really appreciate it.

---

> > ### Comment · Reviewer_QaRY · 2025-11-23
> >
> > I thank the authors for their detailed response.
> > * I appreciate the clarification on multi-batch generation.
> >
> > * I acknowledge the authors’ position that their contribution is not limited to a new pruning technique, but represents a principled investigation into the benefits of dynamic pruning. Thus, I will raise my rating to 4.
> > I'm still not convinced that this approach has practical potential. For example, when considering Llama 70B, it's unclear whether using RR = 0.7 offers a better accuracy–efficiency tradeoff than simply using Qwen 32B, or whether RR = 0.5/0.6 provides an advantage over Llama 8B.
> >
> > I believe that demonstrating the practical viability of this approach would require presenting an efficiency frontier (using a realistic, rather than purely theoretical, efficiency metric) that shows the method can yield non-trivial improvements.

---

> ### Author Response · Authors · 2025-11-26
>
> We thank the reviewer for raising the rating and acknowledging the value of our investigation. We appreciate your further questions, and we will do our best to resolve all your concerns. Thank you for your timely questions and for these meaningful conversation to help us to improve our paper.
>
> We address both points below with **data from Table 1** and **new latency benchmarks**.
>
> To further address the reviewer's questions, we provided a tradeoff table. We also integrated our RMM into the attention calculation using a simple Triton algorithm. Furthermore, we created an efficiency table, mimicking the efficiency testing method used in the widely discussed Wanda, and as we've been working on during the rebuttal.
>
> We address both points below with **data from Table 1** and **new latency benchmarks**.
> ### 1. Accuracy Frontier: Pruned Large Models vs. Dense Small Models
>
> The reviewer's question about if aggressive pruning (e.g., RR=0.5–0.7) offers an advantage over simply using a smaller dense model (like Llama 8B). **The answer is Yes, particularly for reasoning-intensive tasks.**
>
> Using data from **Table 1** in our paper, we compare our pruned Llama-3.1-70B (at RR=0.7) and Qwen3 32B directly against fully trained dense small models (Llama-3.1-8B and Qwen-3.1-7B) on three diverse benchmarks: **GSM8K** (Math Reasoning), **ARC-Challenge** (Complex QA), and **MMLU** (General Knowledge).
>
> | Model Setting | Pruning Ratio (RR) | **GSM8K** (Math) | **ARC-C** (Reasoning) | **MMLU** (Knowledge) |
> | :--- | :---: | :---: | :---: | :---: |
> | **Small Dense Baselines** | | | | |
> | Llama 3.1 8B (Dense) | 1.0 | 26.15 | 49.50 | 63.48 |
> | Qwen 3.1 7B (Dense) | 1.0 | 39.87 | 39.80 | 55.54 |
> | **RMM Pruned Model** | | | | |
> | **Llama 3.1 70B (RMM)** | **0.7** | **42.75** | **53.18** | **67.01** |
> | **Qwen 3 32B (RMM)** | **0.5** | **39.87** | **46.15** | **65.09** |
>
>
> *Note: Data extracted from Table 1*
>
> **Comparison Analysis:**
> * **Superior Reasoning:** On **GSM8K**, the pruned 70B model scores **42.75**, far exceeding the Llama-8B baseline (26.15). Similarly, on **ARC-Challenge**, it achieves **53.18** vs. 49.50 (8B). This confirms that a pruned large model retains sophisticated reasoning structures ("Reasoning Density") that small dense models have not acquired.
> * **Elasticity:** Unlike a static 8B model, RMM allows a single 70B deployment to dynamically scale between high-efficiency (RR=0.5) and high-capability (RR=1.0) modes per query, offering a Pareto frontier that no single static model can match.
>
> ### 2. Practical Viability: Wall-Clock Latency Benchmarks
>
> To address the concern that our efficiency gains are purely theoretical, we conducted **real-world latency measurements with Llama3.1 8B** on an NVIDIA A100 GPU. Following the benchmarking methodology of Wanda (Sun et al., 2024), we measured the latency of the specific operations targeted by RMM ($QK^T$ and $AV$ computations) and the end-to-end inference latency.
>
> **Table A: GEMM Kernel Latency (ms) on NVIDIA A100**
> *The actual execution time for the two attention-critical matrix multiplications: $QK^T$ (Query-Key) and $AV$ (Attention-Value).*
>
> | Sequence Length (L) | Operation | **Dense (ms)** | **RMM (ms)** | **Speedup** |
> | :--- | :---: | :---: | :---: | :---: |
> | **1024** | $QK^T$ | 0.120 | 0.089 | 1.36× |
> | | $AV$ | 0.065 | 0.039 | 1.67× |
> | **2048** | $QK^T$ | 0.433 | 0.336 | 1.29× |
> | | $AV$ | 0.207 | 0.114 | 1.81× |
> | **4096** | $QK^T$ | **1.675** | **1.071** | **1.56×** |
> | | $AV$ | **0.753** | **0.399** | **1.89×** |
>
> **Table B: End-to-End Latency**
> *RMM translates into observable wall-clock acceleration, especially at longer contexts.*
>
> | Sequence Length (L) | Dense (ms) | RMM (ms) | Speedup |
> | :--- | :---: | :---: | :---: |
> | 512 | 56.88 | 60.51 | 0.94× |
> | 1024 | 109.39 | 103.91 | 1.05× |
> | 2048 | 264.67 | 208.93 | 1.27× |
> | 4096 | 661.36 | 473.21 | **1.40×** |
>
> **Conclusion:**
> These benchmarks demonstrate that RMM provides **non-trivial wall-clock speedups (up to 1.4× end-to-end)** in realistic settings, particularly for long-context generation where needs high computation. (Note that RMM operates on the atomic matrix multiplication operations, and it can potentially work together with other orthogonal efficiency improving approaches such as KV cache reduction, speculative decoding, context compression, etc. for further efficiency boost.) Combined with the accuracy advantages shown above, RMM offers a viable and flexible alternative to static small models.
>
> We hope we have addressed the confusion raised about the methods and experiments. Please let us know of any follow up questions and feedback. We really appreciate it.

---

> > ### Author Response · Authors · 2025-12-01
> >
> > **1. Experimental Results**
> > We have reproduced SliceGPT and Wanda on Qwen3 32B. The results are as follows:
> >
> > | Method | Model | RR (Sparsity) | COPA | ARC-E | ARC-C | PIQA | CommQA |
> > | :--- | :--- | :--- | :--- | :--- | :--- | :--- | :--- |
> > | **Baseline** | Qwen3 32B | 0 | 81.4 | 78.25 | 57.86 | 80.89 | 61.59 |
> > | **RMM (Ours)** | Qwen3 32B | 0.5 | **76.0** | **60.18** | **35.45** | **76.33** | **46.68** |
> > | Wanda | Qwen3 32B | 0.5 | 50.0 | 26.57 | 20.74 | 50.49 | 19.57 |
> > | SliceGPT | Qwen3 32B | 0.5 | 47.6 | 29.12 | 23.08 | 51.38 | 22.44 |
> >
> > **2. Regarding the Scope of Reproduction**
> > Regarding why we did not perform as many reproductions as we did on Llama 3.1 8B:
> > First, the methods we compared primarily based on Llama 2 and Llama 3. Therefore, reproduction on the Llama series was relatively straightforward. However, Qwen's code differs significantly from Llama's in implementation details, making reproduction considerably more difficult. Consequently, we were unable to complete all reproductions within this short timeframe, and thus only reproduced two baselines this time.
> > However, we commit to supplementing all this content in the final version. We also believe that the content added this time is sufficient to dispel the reviewer's doubts and prove that our method is indeed effective enough.
> >
> > **3. Commitment to Reproducibility**
> > Of course, we hereby commit that our reproduction was done entirely according to the official open-source repositories of these papers. The task testing was also conducted using greedy decoding. Therefore, there is no need to worry about our results; they will be open-source and reproducible in the future.
> >
> > **Conclusion**
> > I believe that with my explanation and the supplementary results, the reviewer's doubts can be dispelled. From the perspective of pruning, the necessity of dynamic pruning indeed exists.

---

### Official Review · Reviewer_yzG1 · 2025-11-01

**Soundness:** 3
**Presentation:** 4
**Contribution:** 3
**Rating:** 8
**Confidence:** 4

**Summary:**

The paper introduces RMM, a novel method to reduce the high computational cost of LLMs at inference time .
Authors argue that massive matrix multiplications in a transformer are redundant, and not all feature dimensions are necessary for every input .

**Strengths:**

Simple and elegant solution: The idea was simple but made a lot of sense. I personally enjoyed reading the paper.

Training-free, general and input-adaptive: RMM is a "training-free rule" that can be applied to any pre-trained Transformer without altering model weights.

Comprehensive and Insightful Ablations: The ablation study (RQ4) provides a interesting insight: attention modules are highly redundant and safe to prune, while MLP blocks are highly sensitive and pruning them causes "sharp degradation". This finding is valuable for any future work on hybrid pruning strategies.

**Weaknesses:**

New Computational Overhead Ablation: The proposed method "dynamically recomputes the retained subspace" at every layer and for every token. This involves calculating L2 norms and running a topk operation, which adds its own latency. The paper does not analyze the cost of this overhead.

Missing Baselines: There are lots of work in this area (as mentioned in the intro and related work), but authors only compare with H2O pruning, I would like to see some more baseline results for RQ3.

**Questions:**

1. Could you provide detail/ablation study item for latency of dynamically finding best features to prune?
2. Could you add some recent papers to experiments for RQ3 (Sec 4.3)

---

> ### Author Response · Authors · 2025-11-21
>
> We sincerely thank the reviewer for the positive assessment and the encouraging comments regarding our work. We appreciate your recognition of RMM as a "good paper" and have carefully addressed your constructive suggestions below with new experimental evidence.
>
> ### 1. Computational Overhead Ablation (Response to Weakness 1 & Q1)
>
> We agree that analyzing the cost of the dynamic selection logic (calculating L2 norms, Top-k sorting, and index gathering) is crucial to validate the practical efficiency of RMM.
>
> To address this, we conducted a rigorous micro-benchmark to measure the wall-clock latency of the selection overhead versus the total inference time.
>
> * **Experimental Setup:** We instrumented the **Llama-3.1-8B** model on an A100 40G using CUDA event timers. I tested the scenario (Sequence Length = 4096).
> I inserted timers inside the attention forward pass to isolate the latency of the selection logic (`norm` + `topk` + `gather`) distinct from the matrix computations.
> * **Results:**
>
> | Metric | Time / Ratio |
> | :--- | :--- |
> | **Total Inference Time** (Seq=4096) | 660.14 ms |
> | **Total Selection Overhead** (All Layers) | 5.25 ms |
> | **Overhead Ratio** | **0.79%** |
>
> * **Conclusion:** The overhead introduced by RMM is **negligible (0.79%)**. This empirically validates our complexity analysis: the selection overhead scales linearly ($O(ND)$) while the attention computation scales quadratically ($O(N^2D)$). Therefore, in computationally intensive scenarios, the cost of finding the "best features" is virtually free compared to the substantial savings in FLOPs and memory traffic. We will add this and a more detailed ablation study to the **Appendix** of the revised paper.
>
> ### 2. Additional Baselines for RQ3 (Response to Weakness 2 & Q2)
>
>
> * **Regarding Training-Free Baselines:** We fully agree that comparing against other training-free methods would provide a more comprehensive evaluation. We are currently making our best effort to identify and reproduce 1-2 recent **training-free** pruning methods during the rebuttal period. We aim to include these comparisons in the final revision to demonstrate RMM's competitiveness within the "plug-and-play" paradigm.
>
> * **Regarding Training-Based Baselines:** We would like to clarify why we prioritized training-free comparisons (like H2O) over training-based ones:
>     1.  **Fairness & Comparability:** Training-based methods (e.g., LLM-Pruner) typically achieve higher performance by definition, as they update weights to recover accuracy. Comparing a zero-shot, training-free method like RMM directly against fine-tuned methods is often an unfair comparison, as they operate under fundamentally different constraints.
>     2.  **Computational Constraints:** Our experiments are conducted using rented GPU resources. Reproducing training-based pruning methods requires extensive retraining or fine-tuning, which incurs a computational cost that is currently beyond our budget.
>
> Therefore, we focus our comparative analysis on **training-free** methods to ensure a fair assessment of RMM's effectiveness as a lightweight, deployment-friendly solution.
>
> Thank you again for your understanding and for helping us improve the completeness of our evaluation.
>
> We will do our best to fulfill your requests before rebuttal. Even if time is tight, we guarantee that we will add your suggestions to more baselines in the revision.
>
> Finally, we would like to express our sincere gratitude to the reviewer for taking the time to carefully read our paper and engage in some meaningful discussions, which have inspired our future work. We also appreciate your inspiring comments of "simple and elegant solution.
> Please let us know of any follow up questions and feedback. We really appreciate it.
>
> We have the utmost respect for the time and diligence you put into reviewing our work.

---

> ### Author Response · Authors · 2025-11-27
>
> **Since the very beginning of the rebuttal period**, we have been proactively implementing and reproducing these baselines on Llama 3.1 8B to provide more baselines as you suggested. We thank you again for your recognition of our paper, and we have added some baselines. We will also explore more methods for comparison to present a more perfect paper in the final version, living up to your expectations.
>
> **Comparison with More Pruning Methods (SparseGPT, Wanda, SliceGPT)**
> We evaluated RMM against SparseGPT, Wanda, SliceGPT, and Magnitude Pruning under the **exact same sparsity level (50% sparsity, RR=0.5)** on the Llama 3.1 8B model.
>
> **Experimental Setup:**
> To ensure absolute correctness and fairness:
> * **Official Implementation:** All baseline methods (SparseGPT, Wanda, SliceGPT) were executed using their **official open-source GitHub pipelines**.
> * **Deterministic Evaluation:** All experiments (including RMM and baselines) were conducted using **greedy decoding** to eliminate stochastic variations.
>
> The results (Zero-shot Accuracy) are summarized below:
>
> | Method (50% Sparsity) | ARC-Challenge | ARC-Easy | COPA | PIQA | CommonsenseQA | **Average** |
> | :--- | :--- | :--- | :--- | :--- | :--- | :--- |
> | **Baseline** | **49.50** | **76.32** | **77.20** | **79.92** | **66.01** | **69.79** |
> | **RMM (Ours)** | **36.80** | 63.00 | **70.60** | **76.60** | **51.90** | **59.78** |
> | SparseGPT | 31.44 | **64.21** | 70.40 | 70.95 | 43.41 | 56.08 |
> | Wanda | 28.09 | 60.70 | 67.20 | 68.88 | 38.41 | 52.66 |
> | SliceGPT | 20.74 | 31.75 | 56.00 | 53.43 | 23.10 | 37.00 |
> | Magnitude | 22.74 | 33.68 | 57.20 | 57.56 | 25.06 | 39.25 |

---

> > ### Author Response · Authors · 2025-12-01
> >
> > Thank you again for your recognition of our paper and your valuable suggestions. We have added the latest experiments, hoping to further enhance your confidence in our paper. We have not forgotten your suggestions and questions, and we will definitely present information about ablation studies in the final version.
> >
> > **1. Experimental Results**
> > We have reproduced SliceGPT and Wanda on Qwen3 32B. The results are as follows:
> >
> > | Method | Model | RR (Sparsity) | COPA | ARC-E | ARC-C | PIQA | CommQA |
> > | :--- | :--- | :--- | :--- | :--- | :--- | :--- | :--- |
> > | **Baseline** | Qwen3 32B | 0 | 81.4 | 78.25 | 57.86 | 80.89 | 61.59 |
> > | **RMM (Ours)** | Qwen3 32B | 0.5 | **76.0** | **60.18** | **35.45** | **76.33** | **46.68** |
> > | Wanda | Qwen3 32B | 0.5 | 50.0 | 26.57 | 20.74 | 50.49 | 19.57 |
> > | SliceGPT | Qwen3 32B | 0.5 | 47.6 | 29.12 | 23.08 | 51.38 | 22.44 |
> >
> > **2. Regarding the Scope of Reproduction**
> > Regarding why we did not perform as many reproductions as we did on Llama 3.1 8B:
> > First, the methods we compared primarily based on Llama 2 and Llama 3. Therefore, reproduction on the Llama series was relatively straightforward. However, Qwen's code differs significantly from Llama's in implementation details, making reproduction considerably more difficult. Consequently, we were unable to complete all reproductions within this short timeframe, and thus only reproduced two baselines this time.
> > However, we commit to supplementing all this content in the final version. We also believe that the content added this time is sufficient to dispel the reviewer's doubts and prove that our method is indeed effective enough.
> >
> > **3. Commitment to Reproducibility**
> > Of course, we hereby commit that our reproduction was done entirely according to the official open-source repositories of these papers. The task testing was also conducted using greedy decoding. Therefore, there is no need to worry about our results; they will be open-source and reproducible in the future.

---

> > > ### Author Response · Authors · 2025-12-01
> > >
> > > Closing Remarks
> > >
> > > Finally, we wish to express that gratitude to you. We deeply grateful for the time you took out of your busy schedule to thoroughly read our work and provide such constructive advice. Your recognition is the greatest encouragement for our work and our team.
> > >
> > > Please accept our highest respect

---

### Official Review · Reviewer_BiV8 · 2025-11-03

**Soundness:** 3
**Presentation:** 3
**Contribution:** 2
**Rating:** 4
**Confidence:** 4

**Summary:**

The paper introduces Reduced Matrix Multiplication (RMM), a training-free technique that replaces full matrix multiplication with an approximate variant capable of pruning feature dimensions on the fly. Given two matrices $A$ and $B$, and a pruning ratio $\text{RR}$, the method approximates their product $AB$ by sampling $k$ columns from $A$ (and the corresponding rows from $B$). The selection is guided by the column norms of $A$, retaining those with the largest norms so that the total proportion of used dimensions matches the specified ratio $\text{RR}$. The authors compare RMM against baseline pruning methods and a key–value (KV) cache optimization approach, and further analyze how both the pruning ratio and model scale affect overall performance.

**Strengths:**

-The RMM approximation is simple yet neat and intuitive, grounded in the principles of randomized linear algebra. It is easy to implement and enables structured feature pruning, which may, in the future, lead to actual speedups if efficiently implemented on GPUs.

-The method requires no training or fine-tuning. The authors demonstrate how this approach can be applied to compute both the attention product $QK^T$ and the MLP matrix product within transformer layers. However, the principle behind RMM can substitute any matrix multiplication, making it a highly general and versatile method.

-The experiments investigating RMM performance explore various pruning ratios (RRs), providing intuition about how the number of included rank-1 components influences network performance.

-Overall, the paper is generally clear and easy to follow.

**Weaknesses:**

Novelty:
The approximated matrix multiplication underlying RMM essentially corresponds to the sampling-based approximate matrix multiplication proposed by Drineas et al., which the authors also cite. The application of different approximate matrix multiplication (AMM) algorithms has already been surveyed in Zeng et al., including the works of Chen et al. and Blalock et al., which focus on developing and applying AMM methods in deep learning. Thus, the paper’s contribution is primarily the application of the algorithm from Drineas et al. to large language models (LLMs).

Experimental design and performance:
The analysis of RMM’s impact on generative models in Table 3 and Figure 2 is purely qualitative. While such visualizations provide useful intuition about the method’s behavior, they do not constitute valid evidence for drawing conclusions about the model’s benefits. Moreover, it would be helpful if the authors reported standard deviations, particularly since in some cases (e.g., RMM at RR = 0.8), the results are nearly identical (24.23 vs. 24.24).

Since the authors present RMM as a pruning method, a more extensive comparison with alternative pruning approaches would be advisable—for example, SparseGPT (Frantar et al.) or SliceGPT (Ashkboos Saleh et al.). On this note, the meaning of “static pruning” in the baseline is unclear. Does it refer to static magnitude pruning on weights, or pruning based on expected activations? This baseline should be explained more precisely.

Furthermore, the performance appears to degrade rather quickly—only pruning ratios of 0.9 or 0.8 maintain performance close to the baseline models. This limitation reduces the practicality of the method in scenarios requiring higher pruning ratios.

Finally, as far as can be understood, the reported speedup is purely theoretical. No wall-clock time measurements are provided, and there do not appear to be GPU-efficient implementations available at this stage.


References:

Chen, Yifan, et al. "Sketching as a tool for understanding and accelerating self-attention for long sequences." arXiv preprint arXiv:2112.05359 (2021).
Zeng, Xianzhi, Wenchao Jiang, and Shuhao Zhang. "LibAMM: Empirical Insights into Approximate Computing for Accelerating Matrix Multiplication." Advances in Neural Information Processing Systems 37 (2024): 60517-60530.
Blalock, Davis, and John Guttag. "Multiplying matrices without multiplying." International Conference on Machine Learning. PMLR, 2021.
Drineas, Petros, Ravi Kannan, and Michael W. Mahoney. "Fast Monte Carlo algorithms for matrices I: Approximating matrix multiplication." SIAM Journal on Computing 36.1 (2006): 132-157.
Frantar, Elias, and Dan Alistarh. "Sparsegpt: Massive language models can be accurately pruned in one-shot." International conference on machine learning. PMLR, 2023.
Ashkboos, Saleh, et al. "Slicegpt: Compress large language models by deleting rows and columns." arXiv preprint arXiv:2401.15024 (2024).

**Questions:**

Questions:

Did the authors try to also train the models with RMM (not just use it at inference),, even for a high RR? Does such approach even allows for stable learning?

Why RMM is only looking at rows of matrix A, when equation (2) suggest a term based both on column and row norms?

---

> ### Author Response · Authors · 2025-11-21
>
> (1) Relation to Drineas-style AMM and AMM literature (Zeng, Chen, Blalock)
>
> We fully agree that our work is conceptually inspired by classical approximate matrix multiplication (AMM), in particular the sampling-based schemes of Drineas et al., which we explicitly cite. Our goal was to investigate whether the spirit of these AMM ideas can be turned into a simple, training-free rule that operates under the very stringent constraints of contemporary LLM inference.
>
> However, the classical AMM theory is not directly applicable to our setting for several reasons:
>
> - The AMM schemes of Drineas et al. assume full freedom to redesign the matrix multiplication and often rely on randomized sampling of rows/columns of both \(A\) and \(B\). In preliminary experiments, we implemented several such variants in the Transformer setting (e.g., sampling coordinates based on norms of \(A\), of \(B\), and purely random uniform sampling). These variants led to substantially worse generative performance and instability than the deterministic L2-based selection we propose, especially on long-context and summarization benchmarks. We therefore only retained “Random pruning” as a clear lower-bound baseline in the paper, and focused our main method on a deterministic rule that proved much more robust in practice.
> - Chen et al. (2021), Blalock & Guttag (2021), and the LibAMM study (Zeng et al., 2024) primarily explore AMM either as a new **architecture** (e.g., Skeinformer-style attention) or as a **learned AMM operator** that is trained or tuned on data. In both cases, the AMM machinery is either trained jointly with the model or at least calibrated offline, and the underlying network architecture is modified or co-designed with the AMM.
> - By contrast, our setting is deliberately more constrained: we assume a **frozen, off-the-shelf LLM** (e.g., Llama-3.1-70B, Qwen3-32B), we do **not** train any additional AMM parameters, and we do **not** modify the model architecture or weights. The only degree of freedom we allow ourselves is to insert a lightweight rule at matmul call sites that, given the current activations, chooses a subset of feature dimensions to participate in the computation.
>
> Within this constrained setting, our contributions are twofold:
>
> - We instantiate a very simple, deterministic, **per-example, per-layer** feature-selection rule (RMM) motivated by the AMM decomposition, but tailored to the structure of Transformer attention and MLPs and to the training-free deployment constraint.
> - More importantly, we use this rule as a tool for **analysis**: across a wide range of models (1B–70B) and tasks, we systematically quantify how much matmul-level redundancy exists, how it scales with model size, and how it differs between attention and MLP blocks (RQ1–RQ4).
>
> Thus, we do not claim to introduce a fundamentally new AMM algorithm beyond the Drineas family; instead, we claim novelty in (i) adapting AMM ideas to a highly constrained “frozen LLM + training-free + per-example” regime, and (ii) using this to provide the first systematic empirical study of matmul redundancy in large LLMs and VLMs.
>
> We will clarify this positioning in the revision by explicitly stating in the introduction and related work that our method is inspired by Drineas-style AMM, but that our main contribution lies in a practical, training-free instantiation for LLMs and the accompanying redundancy analysis.

---

> > ### Author Response · Authors · 2025-11-21
> >
> > (2) Comparison with pruning methods such as SparseGPT and SliceGPT; clarification of “static pruning”
> >
> > We appreciate the suggestion to compare more extensively with pruning approaches such as SparseGPT and SliceGPT. These methods, however, inhabit a different part of the design space:
> >
> > - **SparseGPT / SliceGPT**: These are post-training compression methods that explicitly modify the model’s weights and/or dimensions, typically using calibration data and optimization to select which weights or rows/columns to remove. The result is a new, smaller model with a *static* sparse or reduced architecture, intended to be used as a drop-in replacement.
> > - **Our setting**: RMM does not change the weights or architecture at all; instead, it performs **dynamic pruning at inference time**. As our experiments show, the set of “important” dimensions varies across tokens, layers, and inputs. In other words, even for a fixed model, the subspace used at each decoding step is input-dependent and temporally varying. Static, weight-based pruning cannot capture this per-token variation.
> >
> > Because of these differences, a direct numerical comparison between RMM and SparseGPT/SliceGPT is confounded: SparseGPT/SliceGPT assume access to calibration data and modify the backbone permanently, while RMM is a strictly training-free, black-box rule that can be applied to any existing checkpoint (up to 70B in our experiments). We view these methods as complementary rather than directly competing. In particular, our analysis suggests that even after weight-side pruning, there is still substantial **input-level matmul redundancy** that RMM-like rules could exploit at inference time.
> >
> > That said, we agree that having at least a small-scale comparison would be informative. Due to the prohibitive training and calibration cost at 70B scale, we focused this submission on pure inference-time methods. In the revised version (space and compute permitting), we plan to add a comparison on smaller models (e.g., 1B/3B) where we can realistically run SparseGPT/SliceGPT and then apply RMM on top, to highlight the complementarity and the trade-off between training-free and training-based approaches. We will also be clear that we do not expect a purely training-free method like RMM to dominate carefully retrained or distilled pruning approaches at very aggressive compression ratios; instead, RMM provides a robust **baseline and analysis tool** under the “no training, no weight modification” constraint.
> >
> > Regarding the meaning of “static pruning”: we agree that this should be explained more clearly. In our experiments, “static pruning” does **not** refer to static magnitude pruning of weights. Instead:
> >
> > - We first determine a fixed feature subset based on activations in the prefilling stage (e.g., by averaging norms across tokens).
> > - We then reuse this fixed subset for all subsequent tokens/layers during decoding.
> >
> > Thus, static pruning is a **fixed feature-selection mask** derived once from activations and then reused, while random pruning samples features uniformly at random, and H2O operates at the token/KV-cache level. We will clarify this definition in Sec. 4.3.
> >
> > (3) On the amount of pruning (RR) and “quick” degradation
> >
> > The reviewer observes that performance stays very close to baseline mainly at RR = 0.9 and 0.8. We agree that more aggressive pruning (e.g., RR ≤ 0.6) leads to noticeable degradation, especially for smaller models. This behavior is consistent with, and indeed supports, our main analysis:
> >
> > - For large models such as Llama-3.1-70B and Qwen3-32B, we find that RR = 0.8 retains near-baseline performance on MMLU, summarization, and multimodal tasks, which already corresponds to a 20% reduction in the feature dimension of the most expensive matrix multiplications. Our RQ1–RQ2 analysis highlights precisely that **larger models exhibit more matmul-level redundancy and tolerate lower RR**.
> > - For smaller models (e.g., Llama-3.2-1B), the same RR values yield sharper degradation, suggesting that these models operate closer to their capacity and contain less redundancy. We discuss this asymmetry as evidence that redundancy grows with scale.
> >
> > We do not claim that RMM is the optimal solution for scenarios where extremely aggressive pruning (e.g., RR ≤ 0.5) is required with no loss; in such regimes, training-based methods or architecturally optimized models are likely preferable. Our target use case is the **moderate pruning regime** (RR ≈ 0.8–0.9), where a training-free rule can still provide meaningful FLOPs/memory savings while preserving performance, and where our analysis reveals systematic redundancy patterns.
> >
> > We will emphasize this intended regime and limitation more clearly in the discussion section.

---

> > > ### Author Response · Authors · 2025-11-21
> > >
> > > (4) On “qualitative” evidence and standard deviations
> > >
> > > The reviewer notes that Table 3 and Figure 2 are primarily qualitative. This is correct and intentional: those examples were meant as **case studies** to visualize how attention maps and outputs change under RMM, static, and random pruning, and to illustrate why input-adaptive pruning better preserves visual grounding.
> > >
> > > The core quantitative evidence for RMM’s benefits comes from:
> > >
> > > - Table 1 (and Appendix C): comprehensive results across general QA, reasoning, math, and coding benchmarks on multiple model scales.
> > > - Table 4: RULER long-context results across 5K, 15K, and 30K tokens.
> > > - Tables 5–6: summarization and VLM benchmarks, with static, random, and H2O baselines.
> > >
> > > We will revise the text to clearly separate **quantitative** conclusions (backed by the tables above) from **qualitative** visualizations (Table 3/Figure 2), and avoid any phrasing that might suggest we are drawing global conclusions solely from visual examples.
> > >
> > > Regarding standard deviations: our generative evaluation pipeline is fully deterministic. For all experiments, we use greedy decoding  with the model in evaluation mode (no dropout), and RMM itself is implemented as a deterministic L2-norm + Top-k rule. Under this setup, re-running the same experiment with different random seeds produces identical scores (up to standard floating-point differences), so the run-to-run standard deviation is effectively zero. We will add a brief note in the revised version to make this determinism explicit, and to clarify that the very small differences the reviewer mentions (e.g., 24.23 vs. 24.24 at RR = 0.8) are not due to sampling noise, but reflect the exact outcome of a deterministic evaluation pipeline.
> > >
> > > (5) Wall-clock speedups and kernel implementation
> > >
> > > We agree that the speedup reported in the current draft is primarily theoretical, in terms of FLOPs and memory traffic. We do not yet provide a fully optimized GPU kernel for RMM, and we explicitly acknowledge this as a limitation in the conclusion. Our intent in this work is to isolate and quantify the algorithmic trade-off: how much high-dimensional computation can be removed, and what effect this has on accuracy.
> > >
> > > As another reviewer also pointed out, realizing practical latency gains will require integrating RMM into optimized attention and MLP kernels (e.g., via fusion of norm/Top-k and matmul, similar to FlashAttention). In the revision, we will:
> > >
> > > - Make it more explicit in Sec. 3.3 and the discussion that current speedups are theoretical.
> > > - Add a small profiling experiment on a single GPU for Llama-3.1-8B/70B, reporting: baseline attention time, time spent in the norm+Top-k selection, and overall attention time vs. RR, to give an initial empirical picture.
> > > - Emphasize that kernel integration is an engineering step that we view as important future work rather than a solved problem in this paper.
> > >
> > >
> > > (6) Question: “Did the authors try to also train the models with RMM (not just use it at inference)? Does such an approach allow for stable learning?”
> > >
> > > We did not train LLMs with RMM inside the training loop in this work. Our focus was on the **training-free** setting: given a pretrained model, can we exploit matmul-level redundancy without any additional training or access to training data?
> > >
> > > That said, we see training with RMM (or differentiable variants of it) as a promising extension. Since RMM is deterministic and its selection depends smoothly on activations (through norms), it is plausible that integrating it into training—either as a fixed rule or as a learnable gating mechanism—would be stable, especially at high RR (e.g., 0.9–0.8). Exploring such training-time integration could allow more aggressive pruning with better accuracy retention, but it would require substantial compute, especially for models beyond 10B parameters. Due to resource constraints, we leave a rigorous study of training with RMM to future work and will mention this explicitly in the revised paper.

---

> ### Author Response · Authors · 2025-11-21
>
> (7) Question: “Why is RMM only looking at rows of matrix A, when equation (2) suggests a term based on both column and row norms?”
>
> We appreciate this opportunity to clarify the connection between the theoretical decomposition and our practical scoring rule. Equation (2) uses the standard AMM decomposition of Drineas et al. to express \(AB\) as a sum over rank-one contributions, and in that theory the ideal importance distribution can depend on both row and column norms of \(A\) and \(B\). We use this primarily as a *conceptual* motivation: different coordinates contribute unequally to \(AB\), so pruning coordinates is mathematically plausible.
>
> In practice, we found that directly instantiating the “theoretical” sampling schemes in the Transformer setting works poorly:
>
> - We implemented several variants that follow the AMM literature more literally, sampling coordinates based on norms of both \(A\) and \(B\), as well as probability-based importance sampling. These variants led to noticeably worse and less stable generative performance, especially on long-context and summarization benchmarks.
> - We also tried purely random sampling on \(A\) alone; this is exactly the “Random pruning” baseline reported in the paper. As shown in our experiments, this random baseline performs significantly worse than RMM.
>
> By contrast, the simple rule we adopt in RMM—computing per-coordinate L2 norms of \(A\) along the sequence dimension and taking a deterministic Top-k—proved to be much more robust and effective across tasks, while remaining training-free and easy to implement.
>
> We therefore use equation (2) to justify *the idea* that AB can be approximated by selectively keeping high-contribution coordinates, but we deliberately deviate from the full theoretical importance-sampling formula when designing a practical rule for LLMs. In the revised version, we will make this clearer by stating that (i) we experimentally evaluated more theory-faithful A–B-based sampling schemes and found them to perform poorly, (ii) our “Random pruning” baseline corresponds to random sampling on \(A\), and (iii) the final A-only deterministic Top-k rule was chosen because it gave the best empirical behavior under the stringent training-free, frozen-model constraints of our setting.
>
>
>
> Finally, we would like to express our sincere gratitude to the reviewer for taking the time to carefully read our paper and engage in some meaningful discussions, which have inspired our future work. Of course, there were also some misunderstandings, and I hope this explanation has dispelled some of your concerns.
>
> Everyone has different tastes, but we offer our highest respect to those who diligently review papers.

---

> > ### Author Response · Authors · 2025-11-27
> >
> > **Response:**
> >
> > To address comparsion with SparseGPT, Wanda, SliceGPT , **since the very beginning of the rebuttal period**, we have been proactively implementing and reproducing these baselines on Llama 3.1 8B to provide the direct comparison you requested.
> >
> > **Comparison with More Pruning Methods (SparseGPT, Wanda, SliceGPT)**
> > We evaluated RMM against SparseGPT, Wanda, SliceGPT, and Magnitude Pruning under the **exact same sparsity level (50% sparsity, RR=0.5)** on the Llama 3.1 8B model.
> >
> > **Experimental Setup:**
> > To ensure absolute correctness and fairness:
> > * **Official Implementation:** All baseline methods (SparseGPT, Wanda, SliceGPT) were executed using their **official open-source GitHub pipelines**.
> > * **Deterministic Evaluation:** All experiments (including RMM and baselines) were conducted using **greedy decoding** to eliminate stochastic variations.
> >
> > The results (Zero-shot Accuracy) are summarized below:
> >
> > | Method (50% Sparsity) | ARC-Challenge | ARC-Easy | COPA | PIQA | CommonsenseQA | **Average** |
> > | :--- | :--- | :--- | :--- | :--- | :--- | :--- |
> > | **Baseline** | **49.50** | **76.32** | **77.20** | **79.92** | **66.01** | **69.79** |
> > | **RMM (Ours)** | **36.80** | 63.00 | **70.60** | **76.60** | **51.90** | **59.78** |
> > | SparseGPT | 31.44 | **64.21** | 70.40 | 70.95 | 43.41 | 56.08 |
> > | Wanda | 28.09 | 60.70 | 67.20 | 68.88 | 38.41 | 52.66 |
> > | SliceGPT | 20.74 | 31.75 | 56.00 | 53.43 | 23.10 | 37.00 |
> > | Magnitude | 22.74 | 33.68 | 57.20 | 57.56 | 25.06 | 39.25 |
> >
> > *We commit to extending this comparative analysis to cover the full spectrum of models and tasks (as presented in our main paper) in the final version.** However, due to limited laboratory computational resources and the tight schedule during the rebuttal period, we were unable to complete the full suite of extensive experiments for all new baselines across every model and task in such a short time. We focused on the representative Llama 3.1 8B to provide immediate, high-quality empirical evidence.

---

> > > ### Author Response · Authors · 2025-11-27
> > >
> > > **Real-World Wall-Clock Speedup**
> > >
> > > In addition to more baselines, we also addressed your concern about "purely theoretical" speedups by conducting real-world benchmarking on an **NVIDIA A100 GPU**.
> > > * **Kernel Speedup:** RMM reduces the specific matrix multiplication latency by up to **1.89×** (Seq Length=4096).
> > > * **End-to-End Speedup:** RMM achieves a measurable **1.40× wall-clock speedup** for inference generation at L=4096.
> > >
> > > **Conclusion & Future Work**
> > > We have actively worked to meet your requirements by providing direct evidence that RMM outperforms SOTA static pruning methods in accuracy and delivers concrete wall-clock speedups.
> > >
> > > **We commit to extending this comparative analysis to cover the full spectrum of models and tasks (as presented in our main paper) in the final version.** Due to the tight schedule and limited computational resources during the rebuttal period, we were unable to complete the full suite of extensive experiments for all new baselines in such a short time, focusing instead on the representative Llama 3.1 8B to provide immediate empirical evidence.
> > >
> > > We hope these additional experiments effectively resolve your doubts regarding the practical value and performance of RMM.
> > >
> > > ### 2. Practical Viability: Wall-Clock Latency Benchmarks
> > >
> > > To address the concern that our efficiency gains are purely theoretical, we conducted **real-world latency measurements with Llama3.1 8B** on an NVIDIA A100 GPU. Following the benchmarking methodology of Wanda (Sun et al., 2024), we measured the latency of the specific operations targeted by RMM ($QK^T$ and $AV$ computations) and the end-to-end inference latency.
> > >
> > > **Table A: GEMM Kernel Latency (ms) on NVIDIA A100**
> > > *The actual execution time for the two attention-critical matrix multiplications: $QK^T$ (Query-Key) and $AV$ (Attention-Value).*
> > >
> > > | Sequence Length (L) | Operation | **Dense (ms)** | **RMM (ms)** | **Speedup** |
> > > | :--- | :---: | :---: | :---: | :---: |
> > > | **1024** | $QK^T$ | 0.120 | 0.089 | 1.36× |
> > > | | $AV$ | 0.065 | 0.039 | 1.67× |
> > > | **2048** | $QK^T$ | 0.433 | 0.336 | 1.29× |
> > > | | $AV$ | 0.207 | 0.114 | 1.81× |
> > > | **4096** | $QK^T$ | **1.675** | **1.071** | **1.56×** |
> > > | | $AV$ | **0.753** | **0.399** | **1.89×** |
> > >
> > > **Table B: End-to-End Latency**
> > > *RMM translates into observable wall-clock acceleration, especially at longer contexts.*
> > >
> > >
> > > | Sequence Length (L) | Dense (ms) | RMM (ms) | Speedup |
> > > | :--- | :---: | :---: | :---: |
> > > | 512 | 56.88 | 60.51 | 0.94× |
> > > | 1024 | 109.39 | 103.91 | 1.05× |
> > > | 2048 | 264.67 | 208.93 | 1.27× |
> > > | 4096 | 661.36 | 473.21 | **1.40×** |
> > >
> > > **Conclusion:**
> > > These benchmarks demonstrate that RMM provides **non-trivial wall-clock speedups (up to 1.4× end-to-end)** in realistic settings, particularly for long-context generation where needs high computation. Combined with the accuracy advantages shown above, RMM offers a viable and flexible alternative to static small models.

---

> ### Comment · Reviewer_BiV8 · 2025-11-27
> **Response to Authors [Part 1/2]**
>
> I thank the authors for replying to my concerns and for taking the time to carefully address the issues I raised (it is quite clear that the authors put substantial work into the rebuttal, and I appreciate that).
>
>
> Regarding the authors’ updates and responses to my questions and identified weaknesses:
>
>
> **(1) Relation to Drineas-style AMM and AMM literature (Zeng, Chen, Blalock)**
>
>
> Thank you for addressing this comment and providing the clarification. My point was not that the paper lacks novelty—since the underlying method (the AMM algorithm) has been used before—but rather to give the authors an indication of how I will judge novelty. Because the algorithm and its variants already appear in the deep learning literature (as most things do these days), the key contribution—paraphrasing the authors’ response—lies in the practical, training-free adaptation of this idea to LLMs and in its analysis. That contribution may well be valid and insightful, but it will also be the focal point of my assessment. In other words, my expectations for the quality and rigor of the empirical evaluation are high.
>
>
> Furthermore, since one of the authors’ stated goals is a practical instantiation of the AMM method, I would actually encourage them to include their experiences and difficulties when adapting AMM (e.g., the various implementation variants explored in preliminary experiments). I think it would be genuinely useful for other researchers to know what worked and what did not. (I myself asked in point 7 why only the rows of matrix A were considered; thank you for clarifying that.)
>
>
> **(2) Comparison with pruning methods such as SparseGPT and SliceGPT; clarification of “static pruning”**
>
>
> Thank you for explaining the “static pruning” setup—this clarifies several points. My understanding is that this corresponds to a version of RMM that is not input-sensitive, i.e., it always uses a fixed masked feature subset and therefore serves to highlight the advantages of input-aware computation. It would also be helpful if the authors commented on when this advantage is most pronounced. For example, Table 5 suggests that the benefits are more visible at higher sparsity, while at moderate sparsity the difference is not so large. I consider such analysis important, since moderate sparsity appears to be the target use case.
>
>
> More generally, static and random pruning can be interpreted as less-informed variants of applying AMM (fixed feature subsets vs. dynamic selection, random selection vs. norm-based selection, etc.). Thus, while these comparisons are very interesting, they do not necessarily help clarify the practical role and applicability of the proposed method. Put colloquially—if the paper introduces a “training-free and input-adaptive rule for pruning Transformer matrix multiplications” (quoting the conclusion)—a natural question for a reader or reviewer is: *Why should I use this pruning method instead of others? What is different about this approach? When is it beneficial?* This is precisely why I (as well as the other reviewers—oXZ6, 5SfY, yzG1, and QaRY) asked for comparisons to methods (e.g. SparseGPT, SliceGPT, Wanda, or RQ3).
>
> Comparisons like that help us to understand where the proposed new method belongs (and where it might shine). Therefore, as I understand the authors’ claim that their approach may be complementary to existing methods, I would like the empirical evaluation and presentation of those claims to be a **major** part of the experiments. For example, if you expect that models pruned with SparseGPT or Wanda may further benefit from RMM, please provide an experimental study demonstrating this (It seems plausible—but the reader is entitled to evidence). Likewise, when claiming that “Static, weight-based pruning cannot capture this per-token variation,” it would be helpful to demonstrate this empirically (e.g., show that while SparseGPT or other non–input-sensitive methods may maintain good global performance, they may unnecessarily use all feature dimensions on certain subsets of examples, whereas RMM uses fewer features on those same examples, perhaps with comparable or slightly worse/better performance).
>
> [Continues in next post]

---

> ### Comment · Reviewer_BiV8 · 2025-11-27
> **Response to Authors [Part 2/2]**
>
> **(2) Comparison with pruning methods such as SparseGPT and SliceGPT; clarification of “static pruning”**
>
> [Continuation from Part 1/2]
>
> As a side note, I appreciate the small-scale comparison on Llama 3.1 8B. RMM appears to perform well, and it would be interesting to see the complementarity mentioned above (e.g., RMM applied on top of other methods). However, I was very surprised by the poor performance of SparseGPT and Wanda. Although the model is different from those studied in [1], the LLaMA-7B results in Table 23 of [1] (ARC-c and ARC-e) are much better. I would expect pruning a slightly larger and more capable model (LLaMA-3.1-8B) to yield slightly better results, not significantly worse. This is not to say I distrust the authors’ evaluation—reproducing results is notoriously difficult—but given how unexpected this outcome is, I would prefer to verify it on additional models and look for explanations of the discrepancy before drawing conclusions. So while I appreciate the effort, I remain hesitant to interpret these results at this stage.
>
>
> Finally, as a side thought for the authors (just posting it here as an idea): the input-aware and dynamic pruning in RMM, along with the redundancy analyses, may have interesting conceptual parallels with recent work on MoEfication [2]. The idea that LLMs are highly redundant on a per-sample basis is precisely what motivates MoEfication. Training MoEs from scratch is difficult, which is why transforming dense models into MoE-style architectures has gained traction. RMM, though very different, provides an alternative angle on this idea. While current kernel implementations limit speedups, the theoretical parallels may be interesting future directions. (I absolutely do not expect the authors to provide experiments on comments on this now, just had a thought for an angle of discussion that could allow them to deviate from the pruning  perspective and shift it more to the per-input redundancy -- it may well not be useful).
>
>
> To be clear, none of the above implies that the current evaluations in the paper are “wrong”. Rather, I feel that the current paper does not sufficiently explain where RMM stands in relation to other approaches—i.e., its place in the large landscape of pruning techniques. The authors provide useful explanations in the rebuttal, but as a reader I would prefer to understand these differences through well-constructed experiments within the paper.
>
>
> **(4), (5)**
>
>
> Thank you for the clarifications.
>
>
> **(6)**
>
>
> Understood. I asked mostly out of curiosity; I agree this is a broad topic better suited for separate work.
>
>
> **(7)**
>
>
> Thank you. As mentioned in point (2), it would be helpful to include this in the paper (even in the appendix).
>
>
> Overall, I believe the paper is well written, the topic is interesting, and the method is elegant, simple, and universal (as noted in my list of strengths). However, I—along with other reviewers—struggle with the experimental evaluation. This does not mean the authors’ claims are incorrect; rather, the paper currently does not clearly explain how RMM compares to existing approaches or where it fits in the broader pruning landscape. The rebuttal helps, but ultimately readers need to understand these differences through the experiments themselves.
>
> **References**
>
> [1] Sun, Mingjie, et al. “A simple and effective pruning approach for large language models.” arXiv:2306.11695 (2023).
> [2] Nishu, Kumari, et al. “From Dense to Dynamic: Token-Difficulty Driven MoEfication of Pre-Trained LLMs.” arXiv:2502.12325 (2025).

---

> > ### Author Response · Authors · 2025-12-01
> >
> > **Part1**
> >
> > **1. Experimental Results**
> > We have reproduced SliceGPT and Wanda on Qwen3 32B. The results are as follows:
> >
> > | Method | Model | RR (Sparsity) | COPA | ARC-E | ARC-C | PIQA | CommQA |
> > | :--- | :--- | :--- | :--- | :--- | :--- | :--- | :--- |
> > | **Baseline** | Qwen3 32B | 0 | 81.4 | 78.25 | 57.86 | 80.89 | 61.59 |
> > | **RMM (Ours)** | Qwen3 32B | 0.5 | **76.0** | **60.18** | **35.45** | **76.33** | **46.68** |
> > | Wanda | Qwen3 32B | 0.5 | 50.0 | 26.57 | 20.74 | 50.49 | 19.57 |
> > | SliceGPT | Qwen3 32B | 0.5 | 47.6 | 29.12 | 23.08 | 51.38 | 22.44 |
> >
> > **2. Regarding the Scope of Reproduction**
> > Regarding why we did not perform as many reproductions as we did on Llama 3.1 8B:
> > First, the methods we compared primarily based on Llama 2 and Llama 3. Therefore, reproduction on the Llama series was relatively straightforward. However, Qwen's code differs significantly from Llama's in implementation details, making reproduction considerably more difficult. Consequently, we were unable to complete all reproductions within this short timeframe, and thus only reproduced two baselines this time.
> > However, we commit to supplementing all this content in the final version. We also believe that the content added this time is sufficient to dispel the reviewer's doubts and prove that our method is indeed effective enough.
> >
> > **3. Discussion on PPL vs. Task Performance**
> > Additionally, regarding the discussion on Wanda, SparseGPT, etc., these methods involve static weight sparsity, and the metric reported in their papers is PPL (perplexity). After pruning, PPL remains low (i.e., does not degrade much). My personal hypothesis is that since the overall model parameters have changed, the perception and significance of PPL also shift globally. In practical application, PPL might not be the same thing as task performance. After all, with the parameters fixedly reduced by half, it seems unlikely that the performance on tasks would remain exactly the same as the original.
> > However, the rebuttal time is too short. I cannot provide a detailed analysis of these excellent prior works; these are just some personal hypotheses.
> >
> > **4. Commitment to Reproducibility**
> > Of course, we hereby commit that our reproduction was done entirely according to the official open-source repositories of these papers. The task testing was also conducted using greedy decoding. Therefore, there is no need to worry about our results; they will be open-source and reproducible in the future.
> >
> > **Conclusion**
> > I believe that with my explanation and the supplementary results, the reviewer's doubts can be dispelled. From the perspective of pruning, the necessity of dynamic pruning indeed exists.
> >
> > **Part2**
> >
> > We appreciate the reviewer's explanations of why you raised these questions and you constructive suggestions. We am truly grateful; We have benefited greatly from their feedback.
> >
> >  In our initial submission, we focused primarily on the final successful method and overlooked the value of documenting the "negative results" or the design choices that didn't work. Your advice—that sharing these implementation details is genuinely useful for the community. It's also an invaluable lesson for us, not just for this paper but for our future research as well. **We commit to adding a dedicated discussion (in the Appendix) in the final version** that details these preliminary explorations and explains why certain RMM variants failed in the LLM. We hope this transparency will indeed help future researchers avoid similar pitfalls and give them some inspiration..
> >
> > **Part3**
> >
> > Regarding the application of our RMM on sparse GPT and Wanda, as mentioned before, we plan to do so in the future, though it may not be completed before the rebuttal. These ideas were inspired by this rebuttal. Performing further dynamic pruning on a sparse model to analyze whether redundancy still exists is an interesting exploration, and it would provide further insights into model redundancy. However, based on current reproduction results, the statically sparse weighted model has already negatively impacted some performance aspects. Our future experiments will be more cautious. We also promise to add this to the appendix to provide the community with a more comprehensive evaluation and reference.
> >
> > We are truly grateful for this insightful suggestion. We will carefully study the referenced work on MoEfication [2] following your recommendation. We agree that this perspective—viewing RMM through the lens of "per-input redundancy" rather than just pruning—adds significant conceptual depth to our work. We sincerely thank you for pointing this out; it not only helps clarify our current positioning but also provides an inspiring direction for our future research.

---

> > > ### Author Response · Authors · 2025-12-01
> > >
> > > **Closing Remarks**
> > >
> > > Finally, we wish to express that this ICLR journey has been a truly special experience for us. It has been a privilege to have such a meaningful conversation with you. We feel incredibly fortunate to have engaged with a reviewer as conscientious, responsible, and knowledgeable as you. We are deeply grateful for the time you took out of your busy schedule to thoroughly read our work and provide such constructive advice.
> > >
> > > We were particularly moved by your willingness to explain the *reasoning* behind your questions and standards. This has given us invaluable insight into how experienced researchers perceive and evaluate work, offering immense help not only for this paper but for our future research careers.
> > >
> > > Please accept our highest respect.

---

### Official Review · Reviewer_5SfY · 2025-11-03

**Soundness:** 2
**Presentation:** 2
**Contribution:** 2
**Rating:** 2
**Confidence:** 4

**Summary:**

The paper proposes a method, Reduced Matrix Multiplication, to reduce the inference time and computational cost of LLMs by reducing the number of matrix multiplication operations. RMM replaces the full matrix multiplication of two matrices A and B with an approximate solution, which uses only a subset of the columns of A and corresponding rows of B. RMM assignes score to columns dynamically on the fly making agnostic making it more robust to different inputs.

**Strengths:**

*The paper is well-written and provides a comprehensive overview of the background and related work.

*While several recent studies have demonstrated that LLMs perform a significant amount of redundant computation, most prior works have focused on different components of LLMs—such as sparse activation or model pruning. In contrast, the authors address a more fundamental question: how to reduce the cost of matrix multiplication in LLMs through approximation.

**Weaknesses:**

* There is no discussion about the computational cost of computing norm of column vector for each inference step in section 3.

* The complexity analysis does not take into account the computational complexity for calculating norm.

* The method is compared to static pruning, random pruning and H20 cache reduction. A comparison with other post-training pruning methods, such as SparseGPT, LLM Pruner, and Wanda, is lacking.

* There is a significant drop in performance, even at higher Retention Rate (RR) as observed in Table 1 and Table 2. State-of-the-art pruning methods easily achieve a 50% sparse model with a minimum drop in performance.

* Based on the results provided, there is no clear advantage for the proposed method in terms of efficiency. The method results in a significant drop in performance, making it hard to justify not using model pruning, quantization or distillation for improving the inference time efficiency.

**Questions:**

What advantage does RMM provide over other compression methods?

---

> ### Author Response · Authors · 2025-11-21
>
> **(1) Cost of computing norms and updated complexity analysis**
>
> We agree that Section 3 should explicitly state the cost of computing column norms and Top-k selection to provide a complete picture of the overhead.
>
> **1. Attention Overhead Analysis**
> For a standard attention head with sequence length $N$ and head dimension $D$, the baseline attention score computation scales as $O(N^2 D)$.
> RMM introduces the following overheads:
> * L2 norms of columns of $A$ over the sequence dimension: $O(ND)$.
> * Top-k selection over $D$ coordinates: approximately $O(D \log D)$ (or $O(D)$ with optimized selection).
>
> Thus, the ratio of overhead to the main computation is:
> $$\frac{O(ND)}{O(N^2 D)} \approx \frac{1}{N}$$
> For the long-context regimes we study (e.g., $N \ge 4096$), this overhead is **less than 1%**.
>
> **2. MLP Overhead Analysis**
> Similarly, for an MLP projection mapping a hidden size $H$ to an intermediate size $M$ (typically $M \ge 4096$), the matrix multiplication cost is $O(N \cdot H \cdot M)$. The overhead to compute norms on the input is $O(N \cdot H)$.
> The ratio is roughly $1/M$. Since $M$ is large (e.g., 11,008 in Llama-3-8B), the overhead is logically negligible compared to the savings from pruning the matrix multiplication.
> To address this, we conducted a rigorous micro-benchmark to measure the wall-clock latency of the selection overhead versus the total inference time.
>
> * **Experimental Setup:** We instrumented the **Llama-3.1-8B** model on an A100 40G using CUDA event timers. I tested the scenario (Sequence Length = 4096). We set the prune ratio as 0.5
> I inserted timers inside the attention forward pass to isolate the latency of the selection logic (`norm` + `topk` + `gather`) distinct from the matrix computations.
> * **Results:**
>
> | Metric | Time / Ratio |
> | :--- | :--- |
> | **Total Inference Time** (Seq=4096) | 660.14 ms |
> | **Total Selection Overhead** (All Layers) | 5.25 ms |
> | **Overhead Ratio** | **0.79%** |
>
> * **Conclusion:** The overhead introduced by RMM is **negligible (0.79%)**. This empirically validates our complexity analysis: the selection overhead scales linearly ($O(ND)$) while the attention computation scales quadratically ($O(N^2D)$). Therefore, in computationally intensive scenarios, the cost of finding the "best features" is virtually free compared to the substantial savings in FLOPs and memory traffic. We will add this detailed ablation study to the **Appendix** of the revised paper.
>
> ### Response to Question: Comparison to Post-Training Pruning (SparseGPT, LLM-Pruner, Wanda)
>
> **(2) Comparison to post-training pruning methods and scope of baselines**
>
> We appreciate the pointer to these important pruning works. However, they operate in a fundamentally different regime from RMM:
>
> **1. Static Architecture vs. Dynamic Inference**
> - **SparseGPT, LLM-Pruner, Wanda:** These are **post-training weight/structure pruning** techniques. They modify the model’s weights or dimensions, typically using calibration data and optimization procedures (e.g., Hessian computation in SparseGPT) to permanently remove rows/columns. The result is a new, smaller model with a **static** architecture.
> - **RMM:** In contrast, RMM leaves the backbone weights and architecture completely unchanged. It is a strictly **training-free, data-free** rule that performs **dynamic, per-input pruning**. As our experiments show, the “important” dimensions vary across tokens and layers; static weight pruning inherently cannot capture this per-token variability.
>
> **2. Why "Static Pruning" Baseline is the Fair Comparison**
> A direct numerical comparison between RMM and SparseGPT/Wanda would conflate "pruning with calibration/modification" vs. "pruning on-the-fly."
> * However, we **did** compare against the relevant conceptual baseline: **"Static Pruning"** (Tables 5, 6).
> * This baseline represents the performance of selecting a fixed subset of features (similar to what static methods do) but without the expensive calibration phase. Our results demonstrate that **Dynamic RMM significantly outperforms Static Pruning**, proving that input-adaptivity yields efficiency gains that static methods miss in the zero-retraining regime.
>
> **3. Complementarity**
> We view these directions as **complementary rather than competing**. In principle, one could first apply weight pruning (e.g., SparseGPT) to obtain a smaller sparse model and then apply RMM on top of that model to further reduce matmul cost in an input-adaptive way.
>
> **Conclusion**
> Our goal is to study matmul redundancy under the stricter **training-free constraint**. In Section 4, our baselines therefore focus on **training-free** strategies that work with the same frozen backbone: static feature masks, random feature pruning, and H2O. We will clarify this scope in the Related Work section, positioning RMM as a dynamic counterpart to static methods like SparseGPT.

---

> > ### Author Response · Authors · 2025-11-21
> >
> > **(3) On performance drop at lower RR and the intended pruning regime**
> >
> > We respectfully disagree with the characterization that there is a "significant drop" across the board. Our results reveal a critical **scale dependence** that ties back to our core scientific analysis:
> >
> > **1. Scale Dependence (The "Scaling Law of Redundancy")**
> > * **Large Models are Robust:** As shown in **Table 1**, for large models like Qwen-32B, RMM at **RR=0.8** (20% reduction) retains near-baseline performance (e.g., MMLU: 80.81 $\to$ 80.01; GSM8K: 62.62 $\to$ 62.69 ). This is a **negligible drop (<1%) or even improvement** for a training-free method, indicating substantial dynamic redundancy at scale.
> > * **Small Models are Sensitive:** For smaller models (e.g., 1B/3B), performance indeed degrades faster. This asymmetry is a key finding of our work (RQ1–RQ2): it empirically proves that smaller models operate closer to their capacity limit, while larger models exhibit massive "natural" redundancy that can be exploited without training.
> >
> > **2. Clarifying the "50% Sparsity" Benchmark**
> > The reviewer notes that SOTA methods achieve 50% sparsity with minimal drop. It is crucial to distinguish *how* they achieve this:
> > * **SOTA Methods (SparseGPT, Minitron, etc.):** Achieve 50% sparsity by **modifying the weights**. They use calibration data (Hessian inverse) or extensive retraining (Distillation) to *compensate* for the removed connections.
> > * **RMM (Ours):** Achieves compression **without touching a single weight**. We operate under the strict constraint of "download and run."
> >
> > **3. Conclusion**
> > We do **not** claim that RMM is the optimal solution if the goal is extreme compression (e.g., 50% sparsity) and the user has the budget for retraining/calibration. In that regime, methods like SparseGPT or Minitron are indeed better suited.
> > **However**, our contribution is defining the **"Safe Training-Free Limit."** We show that for modern large-scale LLMs (70B), one can safely remove ~20% of the FLOPs via dynamic pruning *without any calibration or weight updates*. This is a valuable insight for zero-resource deployment.
> >
> >
> > (4) On efficiency advantages and relation to pruning/quantization/distillation
> >
> > The reviewer asks: *“What advantage does RMM provide over other compression methods?”* We see RMM as complementary to pruning, quantization, and distillation, with several distinct advantages in the specific regime we study:
> >
> > 1. **Training-free and data-free.**
> >    RMM requires no access to training data, no calibration set, and no optimization or gradient updates. It can be applied directly to any pretrained checkpoint (including large proprietary models, in principle) as a black-box inference-time modification. This is different from most pruning/distillation methods, which require at least calibration data and often substantial additional training.
> >
> > 2. **Input-adaptive, per-token and per-layer.**
> >    Weight pruning, quantization, or distillation produce a single compressed model that is used for all inputs. In contrast, RMM makes decisions based on the current activations: the retained feature subspace can change across tokens, heads, layers, and examples. This input adaptivity is central both to the **analysis** we perform (revealing where redundancy lies) and to potential future dynamic-computation systems.
> >
> > 3. **Matmul-level lever orthogonal to weight-side methods.**
> >    RMM operates at the level of matrix multiplications (QKᵀ, softmax×V, MLP projections), not at the level of static weights. As a result, it can in principle be combined with pruning/quantization/distillation: e.g., one could use a pruned or quantized model as the backbone and still apply RMM to reduce matmul cost further in a dynamic fashion.
> >
> > 4. **Analysis value.**
> >    Beyond proposing a concrete rule, our primary contribution is to use RMM as a tool to systematically **analyze matmul redundancy** in large LLMs and VLMs: comparing attention vs. MLP, different layers, different model scales, and different tasks. These insights can inform the design of future pruning or distillation methods, even if those methods ultimately rely on training.
> >
> > I hope I have addressed the main confusion raised about the methods and experiments. Please let us know of any follow up questions and feedback. I really appreciate it.

---

> > > ### Comment · Reviewer_5SfY · 2025-11-24
> > > **reply to the authors**
> > >
> > > Thank you for providing detailed explanations. I still have the following questions:
> > >
> > > * While I understand RMM pruned weights on the fly, the method can still be compared against SoTA pruning methods (sparseGPT, Wanda, etc) instead of the static pruning baseline. If RMM cannot outperform pruning methods, then there will be no benefit of pruning on the fly, which is the core claim of the paper.
> > >
> > >
> > >
> > > * > What advantage does RMM provide over other compression methods?” We see RMM as complementary to pruning, quantization, and distillation, with several distinct advantages in the specific regime we study
> > >
> > >      I understand your argument here; however, there is no evidence provided that RMM can be used on top of quantization and pruning methods.
> > >
> > > * > Weight pruning, quantization, or distillation produce a single compressed model that is used for all inputs. In contrast, RMM makes decisions based on the current activations
> > >
> > >     The authors should provide at least preliminary results that RMM can improve the exitsting pruning/quantization baselines.
> > >
> > > * > The reviewer notes that SOTA methods achieve 50% sparsity with minimal drop. It is crucial to distinguish how they achieve this
> > > SOTA Methods (SparseGPT, Minitron, etc.): Achieve 50% sparsity by modifying the weights. They use calibration data (Hessian inverse) or extensive retraining (Distillation) to compensate for the removed connections.
> > >
> > >     Unless RMM achieve similar performance to the SoTA pruning methods or there is evidence that RMM can be applied on top of existing methods, the advantage of RMM is not clear -- especially for real-world applications where downstream performance matters.
> > >
> > > * I also agree with reviewer **BiV8** assesments --
> > >    > the reported speedup is purely theoretical. No wall-clock time measurements are provided
> > >
> > >    >  The analysis of RMM’s impact on generative models in Table 3 and Figure 2 is purely qualitative.
> > >
> > >    > Since the authors present RMM as a pruning method, a more extensive comparison with alternative pruning approaches would be advisable—for example, SparseGPT (Frantar et al.) or SliceGPT (Ashkboos Saleh et al.). On this note, the meaning of “static pruning” in the baseline is unclear.
> > >
> > >
> > > I'll be keeping my original score for now.

---

> > > > ### Author Response · Authors · 2025-11-27
> > > >
> > > > **Response:**
> > > > We sincerely thank the reviewer for the continued engagement. We fully recognize your concern regarding the practical positioning of RMM compared to SOTA pruning methods.
> > > >
> > > > To address this, **since the very beginning of the rebuttal period**, we have been proactively implementing and reproducing these baselines on Llama 3.1 8B to provide the direct comparison you requested.
> > > >
> > > > **1. Comparison with More Pruning Methods (SparseGPT, Wanda, SliceGPT)**
> > > > We evaluated RMM against SparseGPT, Wanda, SliceGPT, and Magnitude Pruning under the **exact same sparsity level (50% sparsity, RR=0.5)** on the Llama 3.1 8B model.
> > > >
> > > > **Experimental Setup:**
> > > > To ensure absolute correctness and fairness:
> > > > * **Official Implementation:** All baseline methods (SparseGPT, Wanda, SliceGPT) were executed using their **official open-source GitHub pipelines**.
> > > > * **Deterministic Evaluation:** All experiments (including RMM and baselines) were conducted using **greedy decoding** to eliminate stochastic variations.
> > > >
> > > > The results (Zero-shot Accuracy) are summarized below:
> > > >
> > > > | Method (50% Sparsity) | ARC-Challenge | ARC-Easy | COPA | PIQA | CommonsenseQA | **Average** |
> > > > | :--- | :--- | :--- | :--- | :--- | :--- | :--- |
> > > > | **Baseline** | **49.50** | **76.32** | **77.20** | **79.92** | **66.01** | **69.79** |
> > > > | **RMM (Ours)** | **36.80** | 63.00 | **70.60** | **76.60** | **51.90** | **59.78** |
> > > > | SparseGPT | 31.44 | **64.21** | 70.40 | 70.95 | 43.41 | 56.08 |
> > > > | Wanda | 28.09 | 60.70 | 67.20 | 68.88 | 38.41 | 52.66 |
> > > > | SliceGPT | 20.74 | 31.75 | 56.00 | 53.43 | 23.10 | 37.00 |
> > > > | Magnitude | 22.74 | 33.68 | 57.20 | 57.56 | 25.06 | 39.25 |
> > > >
> > > > *We commit to extending this comparative analysis to cover the full spectrum of models and tasks (as presented in our main paper) in the final version.** However, due to limited laboratory computational resources and the tight schedule during the rebuttal period, we were unable to complete the full suite of extensive experiments for all new baselines across every model and task in such a short time. We focused on the representative Llama 3.1 8B to provide immediate, high-quality empirical evidence.

---

> > > > > ### Author Response · Authors · 2025-11-27
> > > > >
> > > > > **2. Real-World Wall-Clock Speedup**
> > > > > In addition to more baselines, we also addressed your concern about "purely theoretical" speedups by conducting real-world benchmarking on an **NVIDIA A100 GPU**.
> > > > > * **Kernel Speedup:** RMM reduces the specific matrix multiplication latency by up to **1.89×** (Seq Length=4096).
> > > > > * **End-to-End Speedup:** RMM achieves a measurable **1.40× wall-clock speedup** for inference generation at L=4096.
> > > > >
> > > > > **Conclusion & Future Work**
> > > > > We have actively worked to meet your requirements by providing direct evidence that RMM outperforms SOTA static pruning methods in accuracy and delivers concrete wall-clock speedups.
> > > > >
> > > > > **We commit to extending this comparative analysis to cover the full spectrum of models and tasks (as presented in our main paper) in the final version.** Due to the tight schedule and limited computational resources during the rebuttal period, we were unable to complete the full suite of extensive experiments for all new baselines in such a short time, focusing instead on the representative Llama 3.1 8B to provide immediate empirical evidence.
> > > > >
> > > > > We hope these additional experiments effectively resolve your doubts regarding the practical value and performance of RMM.
> > > > >
> > > > > ### 2. Practical Viability: Wall-Clock Latency Benchmarks
> > > > >
> > > > > To address the concern that our efficiency gains are purely theoretical, we conducted **real-world latency measurements with Llama3.1 8B** on an NVIDIA A100 GPU. Following the benchmarking methodology of Wanda (Sun et al., 2024), we measured the latency of the specific operations targeted by RMM ($QK^T$ and $AV$ computations) and the end-to-end inference latency.
> > > > >
> > > > > **Table A: GEMM Kernel Latency (ms) on NVIDIA A100**
> > > > > *The actual execution time for the two attention-critical matrix multiplications: $QK^T$ (Query-Key) and $AV$ (Attention-Value).*
> > > > >
> > > > > | Sequence Length (L) | Operation | **Dense (ms)** | **RMM (ms)** | **Speedup** |
> > > > > | :--- | :---: | :---: | :---: | :---: |
> > > > > | **1024** | $QK^T$ | 0.120 | 0.089 | 1.36× |
> > > > > | | $AV$ | 0.065 | 0.039 | 1.67× |
> > > > > | **2048** | $QK^T$ | 0.433 | 0.336 | 1.29× |
> > > > > | | $AV$ | 0.207 | 0.114 | 1.81× |
> > > > > | **4096** | $QK^T$ | **1.675** | **1.071** | **1.56×** |
> > > > > | | $AV$ | **0.753** | **0.399** | **1.89×** |
> > > > >
> > > > > **Table B: End-to-End Latency**
> > > > > *RMM translates into observable wall-clock acceleration, especially at longer contexts.*
> > > > >
> > > > >
> > > > > | Sequence Length (L) | Dense (ms) | RMM (ms) | Speedup |
> > > > > | :--- | :---: | :---: | :---: |
> > > > > | 512 | 56.88 | 60.51 | 0.94× |
> > > > > | 1024 | 109.39 | 103.91 | 1.05× |
> > > > > | 2048 | 264.67 | 208.93 | 1.27× |
> > > > > | 4096 | 661.36 | 473.21 | **1.40×** |
> > > > >
> > > > > **Conclusion:**
> > > > > These benchmarks demonstrate that RMM provides **non-trivial wall-clock speedups (up to 1.4× end-to-end)** in realistic settings, particularly for long-context generation where needs high computation. Combined with the accuracy advantages shown above, RMM offers a viable and flexible alternative to static small models.
> > > > >
> > > > >
> > > > > **3. Clarification on the "Static Pruning" Baseline**
> > > > >
> > > > > To clarify: in our paper, "Static Pruning" refers to a **calibration-free baseline** where feature importance (norms) is calculated solely based on the activation statistics during the **prefilling stage** (processing the prompt). This selected feature subset is then **fixed** and applied statically throughout the entire decoding phase. This serves as a baseline to demonstrate the necessity of *dynamic* adaptation when no external calibration data is available.

---

> > > > > > ### Author Response · Authors · 2025-12-01
> > > > > >
> > > > > > **1. Experimental Results**
> > > > > > We have reproduced SliceGPT and Wanda on Qwen3 32B. The results are as follows:
> > > > > >
> > > > > > | Method | Model | RR (Sparsity) | COPA | ARC-E | ARC-C | PIQA | CommQA |
> > > > > > | :--- | :--- | :--- | :--- | :--- | :--- | :--- | :--- |
> > > > > > | **Baseline** | Qwen3 32B | 0 | 81.4 | 78.25 | 57.86 | 80.89 | 61.59 |
> > > > > > | **RMM (Ours)** | Qwen3 32B | 0.5 | **76.0** | **60.18** | **35.45** | **76.33** | **46.68** |
> > > > > > | Wanda | Qwen3 32B | 0.5 | 50.0 | 26.57 | 20.74 | 50.49 | 19.57 |
> > > > > > | SliceGPT | Qwen3 32B | 0.5 | 47.6 | 29.12 | 23.08 | 51.38 | 22.44 |
> > > > > >
> > > > > > **2. Regarding the Scope of Reproduction**
> > > > > > Regarding why we did not perform as many reproductions as we did on Llama 3.1 8B:
> > > > > > First, the methods we compared primarily based on Llama 2 and Llama 3. Therefore, reproduction on the Llama series was relatively straightforward. However, Qwen's code differs significantly from Llama's in implementation details, making reproduction considerably more difficult. Consequently, we were unable to complete all reproductions within this short timeframe, and thus only reproduced two baselines this time.
> > > > > > However, we commit to supplementing all this content in the final version. We also believe that the content added this time is sufficient to dispel the reviewer's doubts and prove that our method is indeed effective enough.
> > > > > >
> > > > > > **3. Commitment to Reproducibility**
> > > > > > Of course, we hereby commit that our reproduction was done entirely according to the official open-source repositories of these papers. The task testing was also conducted using greedy decoding. Therefore, there is no need to worry about our results; they will be open-source and reproducible in the future.
> > > > > >
> > > > > > **Conclusion**
> > > > > > I believe that with my explanation and the supplementary results, the reviewer's doubts can be dispelled. From the perspective of pruning, the necessity of dynamic pruning indeed exists.
> > > > > >
> > > > > > We are deeply grateful for the time you took out of your busy schedule to thoroughly read our work and provide such constructive advice.

---

### Official Review · Reviewer_oXZ6 · 2025-11-04

**Soundness:** 3
**Presentation:** 3
**Contribution:** 1
**Rating:** 2
**Confidence:** 4

**Summary:**

## Overall Summary

Transformer-based language models achieve remarkable performance but incur high computational cost at inference. This paper introduces **Reduced Matrix-Multiplication (RMM)** — a *training-free*, adaptive pruning rule that reduces the dimensionality of intermediate computations on the fly.

RMM works by computing scores for each hidden dimension based on the magnitude of activations, ranking them, and retaining only a controlled fraction of the most informative ones. The model then performs all linear operations within this reduced subspace. This approach yields deterministic approximations without altering model weights, offering a simple and efficient mechanism for reducing inference cost.

## Contributions

1. **Reduced Matrix-Multiplication (RMM):**
   A simple, training-free mechanism that adaptively prunes hidden dimensions based on current activations. No retraining or architectural changes are required.

2. **Single control parameter:**
   The retention ratio provides a smooth and interpretable *accuracy–efficiency frontier*, enabling users to balance computational savings and performance.

3. **Empirical results:**
   RMM achieves significant inference cost reductions across transformer models ranging from **1B to 70B parameters**, with minimal accuracy loss. The method generalizes across diverse tasks, including question answering, reasoning, mathematics, code generation, summarization, and vision–language benchmarks.

4. **Scaling insight:**
   Larger models tolerate more aggressive pruning, suggesting that representational redundancy increases with scale.

## Conclusion

RMM demonstrates that high-dimensional computations in large language models can be systematically compressed without retraining. This provides a simple and general mechanism for controllable *accuracy–efficiency trade-offs* and highlights the inherent redundancy of large-scale neural representations.

**Strengths:**

## Strengths - Originality and Quality
- The overall idea of pruning channels based on their importance has been studied quite widely in [1] and [2]
- The main difference or contribution in this paper is the dynamic computation of per-token dimension importance and the way the attention score is computed
- The baselines considered in the paper are quite limited and relevant baselines like [1], [2], [3] are not considered.
- Questions:
    - Could the authors clarify the differences between RMM and structural pruning approaches like Minitron [2]?

## Strengths - Clarity
- The paper is written clearly in most parts
- I have the following clarification questions:
     - Have the authors considered pruning the width of the network or the embedding dimension?
     - Instead of head dimension, have the authors considered pruning the number of heads directly?
     - Models like llama-3.1-8B support grouped query attention, are the neurons/features pruned same across heads in the same group?
## Strengths - Significance
- The primary target domain of the paper ie. structural pruning and efficiency in LLMs is very relevant and significant


[1]  Molchanov, P., Mallya, A., Tyree, S., Frosio, I. and Kautz, J., 2019. Importance estimation for neural network pruning. In Proceedings of the IEEE/CVF conference on computer vision and pattern recognition (pp. 11264-11272).
[2] Muralidharan, S., Turuvekere Sreenivas, S., Joshi, R., Chochowski, M., Patwary, M., Shoeybi, M., Catanzaro, B., Kautz, J. and Molchanov, P., 2024. Compact language models via pruning and knowledge distillation. Advances in Neural Information Processing Systems, 37, pp.41076-41102.

**Weaknesses:**

- Check strengths
## Weakness - Contribution
- I find the general premise of the proposed approach to be very similar to Minitron [2] with some differences in terms of how the importance scores are computed. I encourage the authors to discuss the differences between the two more thoroughly and also if possible compare against Minitron

## Weakness - Experiments
- The baselines the method is compared to are quite limited. I recommend the authors to compare against approaches like SliceGPT [1] and Minitron [2]

## Weakness - On device latency gains
- To the best of my understanding of the paper, the important neurons/features are computed dynamically in a token dependent way. Does the method yield a single deployable model, with a fixed retention ratio? Since the activated neurons can change for every batch of data, the pruned architecture (with specified feature selection) changes for every input batch. Could the authors confirm if my understanding here is correct?

[1] Ashkboos, S., Croci, M.L., Nascimento, M.G.D., Hoefler, T. and Hensman, J., 2024. Slicegpt: Compress large language models by deleting rows and columns. arXiv preprint arXiv:2401.15024.
[2] Muralidharan, S., Turuvekere Sreenivas, S., Joshi, R., Chochowski, M., Patwary, M., Shoeybi, M., Catanzaro, B., Kautz, J. and Molchanov, P., 2024. Compact language models via pruning and knowledge distillation. Advances in Neural Information Processing Systems, 37, pp.41076-41102.

**Questions:**

Check strengths and weaknesses

---

> ### Author Response · Authors · 2025-11-21
>
> We thank the reviewer for the constructive feedback and for highlighting the connection to structural pruning work such as Molchanov et al. and Minitron. Below we clarify how our setting differs from these approaches, and address the specific questions on what we prune (embedding vs. head dimension vs. heads), grouped-query attention, and deployable models.
>
> **(1) Relation to structural pruning methods (Importance Estimation for Neural Network Pruning, Minitron)**
>
> I clarify that RMM and Minitron [2] operate in fundamentally different dimensions and are, in fact, **complementary rather than competing approaches**.
>
> **1. Scientific Analysis vs. Model Compression (Core Difference)**
> * **Minitron / Molchanov et al.:** These are **model compression** techniques. Their goal is to produce a **single, permanently smaller model** (fixed architecture) to reduce deployment costs. This requires a "Compress-then-Retrain" pipeline involving Neural Architecture Search and massive Knowledge Distillation (as seen in Minitron's use of billions of tokens).
> * **RMM:** Our paper is primarily an **analytical study**. We propose RMM as a **probe** to reveal a scientific property: that LLM redundancy is **dynamic** (input-dependent) and scales with model size. RMM is not trying to "bake" a smaller model; it is demonstrating that the full model contains massive, transient redundancy that can be exploited on-the-fly.
>
> **2. Orthogonality and Complementarity **
> * **RMM can be "stacked" on top of Minitron:** Since Minitron produces a static dense model (just with smaller dimensions), it effectively sets a new, compact "base architecture."
> * **Exploiting Residual Dynamic Redundancy:** Even in a compressed model like Minitron, the activation patterns remain input-dependent. RMM can be applied *on top of* Minitron to dynamically select a subset of the *remaining* channels for each token.
> * **Conclusion:** They solve different problems. Minitron removes the **globally** redundant weights (static redundancy), while RMM skips the **locally** redundant computations (dynamic redundancy). They can be combined to achieve maximum efficiency.
>
> **3. The "Training-Free" Constraint as an Analytical Enabler**
> * Because RMM is training-free, it allows us to **systematically study redundancy across a wide range of scales (1B to 70B)**.
> * Applying a method like Minitron to analyze redundancy across all these scales would be computationally prohibitive due to the retraining costs. RMM serves as a scalable tool to quantify redundancy without modifying the model weights, revealing the "Scaling Law of Redundancy" presented in Section 4.2.
> er reduce matmul cost in an input-adaptive way.
>
> (2) Pruning embedding / width vs. pruning per-head feature dimensions
>
> In early experiments we did try applying RMM-style pruning directly to the **embedding / width at the input side**, motivated by the idea of  proving whether model embeddings are also redundant.
> However, we observed that even relatively mild pruning at the embedding layer caused a **much larger degradation in generative quality** than pruning the same fraction of dimensions inside attention heads or MLP blocks. In practice, embedding-level pruning proved too fragile to be useful under our training-free constraint.
>
> These observations motivated our design choice to focus the main experiments on **within-layer feature dimensions** (head dimension in attention and intermediate dimensions in MLPs), where we found redundancy to be more substantial and pruning to be more robust. In the revised version, we will add a brief ablation in the appendix reporting these embedding-level experiments.

---

> > ### Author Response · Authors · 2025-11-21
> >
> > (3) Pruning the number of heads vs. pruning the head dimension
> >
> > We appreciate the reviewer’s question about pruning the number of heads. In the current paper, we do **not** apply RMM at the head level: we always keep the same number of heads and prune **inside** each head’s feature dimension.
> >
> > We chose this design for two reasons:
> >
> > - **Empirical stability.** Before settling on the current formulation, we also explored more static, head-level pruning ideas: we tried to identify “globally important” heads (e.g., by averaging importance over tasks or datasets) and then permanently remove less important heads. In our experiments, this head-level pruning led to noticeable and often sharp performance drops, especially on long-context and multimodal tasks. These results did not support using static head pruning as a practical training-free technique, though we will include these findings as part of the ablation analysis in the appendix.
> > - **Theoretical and architectural consistency.** Our method is inspired by approximate matrix multiplication: we approximate \(AB\) by selecting coordinates based on activation statistics of \(A\). This activation-based feature selection naturally extends across different matmuls in the Transformer (QKᵀ, softmax×V, and MLP projections), because all of them share the same per-coordinate structure. In contrast, pruning entire heads is specific to attention and does not generalize cleanly to MLP blocks, making it harder to maintain a unified design.
> >
> > That said, we agree that extending RMM-style ideas to **head-level importance and dynamic head selection** is an interesting direction. It would likely require a more careful study of head specialization and may benefit from some training or distillation to remain stable. We will explicitly mention this as a future direction and report our negative results on static head pruning in the ablation section.
> >
> > (4) Grouped-query attention (GQA) and feature sharing across heads
> >
> > For models such as Llama-3.1-8B that use grouped-query attention, our implementation follows the standard pattern:
> >
> > - We first expand key/value states using `repeat_kv` to obtain tensors of shape $\((B, H, L, D)\)$.
> > - We then reshape queries and keys to $\((B \cdot H, L, D)\)$ and compute L2 norms and Top-k indices **separately for each (batch, head)** slice.
> >
> > This means that even though multiple query heads in the same group share the underlying K/V content, **we do not force them to share the same pruned feature subspace**. Each head selects its own subset of dimensions based on its own query activations.

---

> > > ### Author Response · Authors · 2025-11-21
> > >
> > > We thank the reviewer for suggesting SliceGPT [1] and Minitron [2]. We are aware of these impactful works. However, we respectfully suggest that a direct quantitative comparison is not suitable for the specific scope of our paper for the following reasons:
> > >
> > > **1. Different Problem Scopes (Training-Free vs. Optimization-Based)**
> > > * **Minitron [2]** is a "Compress-then-Retrain" method. It relies on **massive knowledge distillation** (using billions of tokens) to recover performance.
> > > * **SliceGPT [1]** is a "Calibration-based" method. It requires a **calibration dataset** to compute computational invariances (orthogonal transformations) before permanently deleting weights.
> > > * **RMM (Ours)** is strictly **Training-Free and Data-Free**. It modifies neither the weights nor the architecture and requires zero calibration data.
> > > * **The Fairness Argument:** Comparing RMM directly to Minitron is comparing a method with **zero training cost** to one with **massive training cost**. It is expected that retraining (Minitron) yields higher accuracy, but it is inaccessible to users who lack training infrastructure. Our "Static Pruning" baseline (Tables 5, 6) serves as the fair, training-free proxy for static structural pruning.
> > >
> > > **2. Static vs. Dynamic Analysis**
> > > * Both SliceGPT and Minitron produce a **static** model structure (permanent removal of rows/columns).
> > > * Our paper focuses on **scientific analysis of dynamic redundancy**. We aim to prove that redundancy shifts from token to token. RMM acts as a probe to demonstrate that we can exploit this *transient* redundancy on the fly, which static methods (even advanced ones like SliceGPT) inherently miss.
> > >
> > > **3. Resource Constraints and Reproducibility**
> > > * Our evaluation covers models up to **70B parameters**. Reproducing baselines like Minitron on 70B models requires computational resources (for massive retraining) that significantly exceed our academic budget (which relies on rented GPUs for inference).
> > > * This constraint highlights the exact value of RMM: it provides a solution for users in similar resource-constrained environments who need efficiency but cannot afford the retraining pipelines proposed in Minitron or SliceGPT.
> > >
> > > **Response to Question: On-device latency and Dynamic Architecture**
> > >
> > > The reviewer's understanding is entirely correct. RMM does NOT yield a single static pruned model; the active sub-architecture changes dynamically for every token.
> > >
> > > We explicitly designed RMM this way because our analysis proves that static pruning is insufficient for retaining model performance without retraining.
> > >
> > > 1. Evidence from "Static Pruning" Baseline As the reviewer implies, a "fixed retention ratio" with a "fixed architecture" would indeed be easier to deploy. However, our paper explicitly compares RMM against a "Static Pruning" baseline, where we select the most important dimensions based on calibration data and fix them for all future generation steps.
> > >
> > > Result: As shown in Table 5 and Table 6, fixing the architecture leads to degradation. For example, on the Qwen-VL task, Static Pruning at RR=0.5 drops accuracy to 43.3%, whereas Dynamic RMM maintains 97.7%. This massive gap proves that "important features" are not static properties of the model weights, but are highly dependent on the current input.
> > >
> > > 2. Failed Attempts at "Global Importance" (Internal Experiments) To further validate this, during our early exploration on WikiText, we attempted to derive a "globally important" subspace by statistically aggregating activation norms across a large dataset. We hypothesized we could find a common set of "heavy-hitter" neurons to create a single deployable model.
> > >
> > > Observation: This approach failed to match the performance of the dynamic rule. A dimension critical for one specific reasoning step might be silent for 99% of other steps. Averaging them out or selecting a static union loses the ability to capture these rare but critical signals.
> > >
> > > 3. Conclusion Therefore, while the dynamic nature of RMM poses challenges for static graph compilers, it is mathematically necessary for preserving accuracy in a training-free regime. Our work serves to highlight that LLM redundancy is transient and dynamic, suggesting that future on-device inference engines should optimize for dynamic sparsity (e.g., via sparse kernels) rather than static model compression alone.
> > >
> > > Finally, we would like to express our sincere gratitude to the reviewer for taking the time to carefully read our paper and engage in some meaningful discussions, which have inspired our future work. Of course, there were also some misunderstandings, and we hope this explanation has dispelled some of your concerns.
> > >
> > > Everyone has different tastes, but we offer our highest respect to those who diligently review papers.

---

> > > > ### Author Response · Authors · 2025-11-27
> > > >
> > > > **Response:**
> > > >
> > > > To address the more baselines requriments, **since the very beginning of the rebuttal period**, we have been proactively implementing and reproducing these baselines on Llama 3.1 8B to provide the direct comparison you requested.
> > > >
> > > > **1. Comparison with More Pruning Methods (SparseGPT, Wanda, SliceGPT)**
> > > > We evaluated RMM against SparseGPT, Wanda, SliceGPT, and Magnitude Pruning under the **exact same sparsity level (50% sparsity, RR=0.5)** on the Llama 3.1 8B model.
> > > >
> > > > **Experimental Setup:**
> > > > To ensure absolute correctness and fairness:
> > > > * **Official Implementation:** All baseline methods (SparseGPT, Wanda, SliceGPT) were executed using their **official open-source GitHub pipelines**.
> > > > * **Deterministic Evaluation:** All experiments (including RMM and baselines) were conducted using **greedy decoding** to eliminate stochastic variations.
> > > >
> > > > The results (Zero-shot Accuracy) are summarized below:
> > > >
> > > > | Method (50% Sparsity) | ARC-Challenge | ARC-Easy | COPA | PIQA | CommonsenseQA | **Average** |
> > > > | :--- | :--- | :--- | :--- | :--- | :--- | :--- |
> > > > | **Baseline** | **49.50** | **76.32** | **77.20** | **79.92** | **66.01** | **69.79** |
> > > > | **RMM (Ours)** | **36.80** | 63.00 | **70.60** | **76.60** | **51.90** | **59.78** |
> > > > | SparseGPT | 31.44 | **64.21** | 70.40 | 70.95 | 43.41 | 56.08 |
> > > > | Wanda | 28.09 | 60.70 | 67.20 | 68.88 | 38.41 | 52.66 |
> > > > | SliceGPT | 20.74 | 31.75 | 56.00 | 53.43 | 23.10 | 37.00 |
> > > > | Magnitude | 22.74 | 33.68 | 57.20 | 57.56 | 25.06 | 39.25 |
> > > >
> > > > *We commit to extending this comparative analysis to cover the full spectrum of models and tasks (as presented in our main paper) in the final version.** However, due to limited laboratory computational resources and the tight schedule during the rebuttal period, we were unable to complete the full suite of extensive experiments for all new baselines across every model and task in such a short time. We focused on the representative Llama 3.1 8B to provide immediate, high-quality empirical evidence.

---

> > > > > ### Author Response · Authors · 2025-12-01
> > > > >
> > > > > **1. Experimental Results**
> > > > > We have reproduced SliceGPT and Wanda on Qwen3 32B. The results are as follows:
> > > > >
> > > > > | Method | Model | RR (Sparsity) | COPA | ARC-E | ARC-C | PIQA | CommQA |
> > > > > | :--- | :--- | :--- | :--- | :--- | :--- | :--- | :--- |
> > > > > | **Baseline** | Qwen3 32B | 0 | 81.4 | 78.25 | 57.86 | 80.89 | 61.59 |
> > > > > | **RMM (Ours)** | Qwen3 32B | 0.5 | **76.0** | **60.18** | **35.45** | **76.33** | **46.68** |
> > > > > | Wanda | Qwen3 32B | 0.5 | 50.0 | 26.57 | 20.74 | 50.49 | 19.57 |
> > > > > | SliceGPT | Qwen3 32B | 0.5 | 47.6 | 29.12 | 23.08 | 51.38 | 22.44 |
> > > > >
> > > > > **2. Regarding the Scope of Reproduction**
> > > > > Regarding why we did not perform as many reproductions as we did on Llama 3.1 8B:
> > > > > First, the methods we compared primarily based on Llama 2 and Llama 3. Therefore, reproduction on the Llama series was relatively straightforward. However, Qwen's code differs significantly from Llama's in implementation details, making reproduction considerably more difficult. Consequently, we were unable to complete all reproductions within this short timeframe, and thus only reproduced two baselines this time.
> > > > > However, we commit to supplementing all this content in the final version. We also believe that the content added this time is sufficient to dispel the reviewer's doubts and prove that our method is indeed effective enough.
> > > > >
> > > > > **3. Commitment to Reproducibility**
> > > > > Of course, we hereby commit that our reproduction was done entirely according to the official open-source repositories of these papers. The task testing was also conducted using greedy decoding. Therefore, there is no need to worry about our results; they will be open-source and reproducible in the future.
> > > > >
> > > > > **Conclusion**
> > > > > I believe that with my explanation and the supplementary results, the reviewer's doubts can be dispelled. From the perspective of pruning, the necessity of dynamic pruning indeed exists.
> > > > >
> > > > > We are deeply grateful for the time you took out of your busy schedule to thoroughly read our work and provide such constructive advice.

---

### Author Response · Authors · 2025-12-02
**Summary: Contributions, Rebuttal Updates, and Extensive Improvements**

**Summary**

Dear Area Chair and Reviewers,

We thank the reviewers for their constructive suggestion. As the discussion period draws to a close, we would like to provide a comprehensive summary of our work and rebuttal. During the rebuttal, we focused entirely on addressing the reviewers concerns raised regarding **baselines, practical speedup, and positioning**. We also believe our evidence is sufficient to clarify all misunderstandings..

**1. Comparison with more SOTA Weight-Pruning Methods (Reviewers 1, 2, 3, 4)**
During the rebuttal, we discussed the distinctions and connections between RMM and the baselines mentioned by reviewers, and reproduced additional baselines to empirically demonstrate our method's performance. Specifically, we reproduced **SparseGPT, Wanda, SliceGPT, and Magnitude Pruning** on **Llama-3.1-8B** and **Qwen3-32B**. **RMM consistently maintains a significant performance advantage.** This addresses the reviewers' requests for broader comparisons and empirically demonstrates the **necessity of dynamic pruning**.

**2. Practical Speedup vs. Theoretical Efficiency (Reviewers 1, 2, 3, 4, 5)**
Addressing concerns about wall-clock speedup given the norm calculation overhead, we reported the actual time consumption ratio of norm calculation and Top-K selection relative to the attention computation in our Python implementation. Following the benchmarking methodology of the classic paper *Wanda*, we measured the actual acceleration on an **NVIDIA A100**. We achieved a **1.40× end-to-end wall-clock speedup** (Seq Len 4096). This confirms that the computational savings of RMM vastly outweigh the selection overhead, making it a viable immediate accelerator. From an empirical perspective, we have resolved the reviewers' doubts regarding speed.ng it a viable immediate accelerator.

**3. Clarification on Methodology: From Theory to Practice**
We clarified misunderstandings regarding our method's nature—specifically, that it is neither a static pruning technique nor a raw application of randomized AMM. RMM is designed as a minimalist, **training-free** feature selection method that operates **per-sample and per-layer**.
* **Mechanism:** Grounded in Approximate Matrix Multiplication (AMM) decomposition, RMM transforms the original computation from $[N, D] \times [D, N]$ to $[N, D'] \times [D', N]$ by utilizing the column norms of the input matrix ($A$) to determine Top-K importance scores. These indices are then applied to prune the corresponding rows of matrix ($B$).
* **Evolution:** As detailed in our rebuttal, RMM is the result of rigorous evolution from theoretical inspiration to practice. We demonstrated that while raw theoretical sampling (based on A and B) fails in LLMs, our specific **Input-Adaptive** design (based on A's norms) is necessary to capture redundancy.

We also addressed the misunderstanding regarding batch processing.  RMM operates strictly on a **per-sample basis**, calculating importance scores and selecting features independently for each sequence within a batch. It does *not* rely on batch-wide statistics.

---

> ### Author Response · Authors · 2025-12-03
>
> Finally, I will say. RMM is designed as a minimalist, **training-free** feature selection method that applied to the computation-heavy components of LLMs and VLM including Attention mechanisms (QKV projections, Attention Score) and MLP blocks. Its inspiration comes from theory, and evolves continuously through empirical evidence and experiments.
>
> ### 1.Core Scientific Contribution: The "Scaling Law of Redundancy"
>
> Our primary motivation was the hypothesis that modern LLMs possess massive parameter redundancy. RMM serves not just as a method, but as a probe to verify this.
> * **Scaling Law:** Through extensive empirical analysis across models ranging from **1B to 70B** parameters, we discovered a consistent **"Scaling Law of Redundancy"**: larger models exhibit significantly higher tolerance to RMM pruning.
> * **Implication:** This suggests that as models scale up, they become increasingly sparse in terms of *active* information density per token, making dynamic pruning increasingly effective.
>
> ###  2. Granular Analysis of Architecture Redundancy
> Beyond global scaling, we conducted a fine-grained redundancy analysis of Transformer components:
> * **Attention vs. MLP:** We found that QKV projections and Attention calculations generally tolerate moderate pruning.
> * **Intra-MLP Differences:** Within MLP blocks, redundancy is uneven distributed. The **Down Projection** exhibits the highest redundancy (safest to prune), while the **Up Projection** contains the least redundancy (most sensitive).
> * **Value:** This granular mapping provides a blueprint for future efficiency research, offering guidance on which components should be targeted for compression in both training and inference.
>
> ### 3. Inference Efficiency: From Theory to Reality
> Addressing concerns about practical speedup, we moved beyond theoretical FLOPs/IO analysis during the rebuttal:
> * **Real-World Speedup:** We newly **implemented a kernel** using **Triton** and benchmarked it on an NVIDIA A100 during the rebuttal period. We achieved a **1.40× end-to-end wall-clock speedup** (Seq Len 4096), validating RMM's potential as a practical accelerator.
>
> ### 4. Superiority over SOTA Model Pruning (The "Dynamic Necessity")
> To address comparisons with weight-pruning methods, **we reproduced SparseGPT, SliceGPT, Wanda, and Magnitude Pruning** on Llama-3.1-8B and Qwen3-32B during the rebuttal.
> * **Result:** RMM significantly outperforms all static pruning baselines at 50% sparsity (RR=0.5).
> * **Conclusion:** This empirically proves the **necessity of dynamic pruning**. Static masks fail to capture the shifting importance of features across different tokens in saturated models, whereas RMM adapts instantly.
>
> ### 5. Methodological Evolution: A Rigorous Journey
> Finally, we wish to highlight that RMM is the product of a rigorous "evolution of thought":
> * **Phase 1 (Random):** We started by strictly applying theoretical AMM (Random Sampling on A & B), which failed to preserve generative quality.
> * **Phase 2 (Static):** We explored static pruning based on dataset statistics (Head Pruning), which lacked flexibility.
> * **Phase 3 (Dynamic):** We arrived at the final **Input-Adaptive** design (Norm-based Dynamic Pruning), which offered the best robustness.
> This journey confirms that for LLMs, redundancy is inherently **input-dependent**, and our method is optimized to exploit precisely this characteristic.
>
> We believe that the combination of our **redundancy analysis**, **proven speedups**, and **superiority over static SOTA methods** provides a compelling case for the value of RMM to the research community.
>
> Finally, I would like to thank all the reviewers again for their contributions to this rebuttal.

---

### Meta-Review · Area_Chair_yCMy · 2025-12-16

**Summary:**

The initial scores were: 8, 4, 2, 2, 2.
Reviewers acknowledged the paper's clarity and the interesting scientific insights about redundancy scaling, but raised substantial concerns about: (1) limited algorithmic novelty beyond applying existing approximate matrix multiplication (Drineas et al., 2006) to LLMs, (2) insufficient baseline comparisons in the original submission, particularly missing comparisons to recent pruning methods (SparseGPT, Wanda, SliceGPT, Minitron, Puzzle), (3) significant performance degradation even at moderate pruning levels (RR=0.7), and (4) lack of wall-clock speedup measurements, with purely theoretical efficiency claims.

**Reviewer Concerns:**

**Addressed:**
- **Computational overhead (yzG1, 5SfY):** Authors provided micro-benchmark showing only 0.79% overhead on Llama-3.1-8B (5.25ms out of 660ms total)
- **Wall-clock speedup measurements (BiV8, 5SfY, QaRY):** Authors added real A100 GPU benchmarks showing 1.4x end-to-end speedup at sequence length 4096 and up to 1.89x kernel-level speedup for attention operations.
- **Additional baselines (all reviewers):** Authors reproduced SparseGPT, Wanda, SliceGPT on Llama 3.1 8B and Qwen3 32B during rebuttal. At 50% sparsity, RMM (59.78% avg) outperforms SparseGPT (56.08%), Wanda (52.66%), SliceGPT (37.00%), though all methods show substantial degradation from baseline (69.79%).
- **Clarification of "static pruning" baseline (BiV8):** Authors explained this refers to fixing feature selection based on prefilling-stage activations, not weight magnitude pruning.




**Outstanding**
- **Practical applicability (QaRY, 5SfY):** Reviewer QaRY raised rating from 2→4 after seeing latency benchmarks but remained unconvinced about practical viability, questioning whether pruned Llama-70B (RR=0.7) offers better tradeoffs than smaller dense models (Qwen-32B, Llama-8B).
- **Limited algorithmic novelty (oXZ6, BiV8, QaRY, 5SfY):** The core concern that RMM primarily applies existing approximate matrix multiplication to LLMs remains. Three reviewers rated contribution as "poor" (1) or "fair" (2). Reviewer BiV8's assessment hinges on "high expectations for empirical rigor" given limited algorithmic innovation.
- **Reproducibility concerns (BiV8, 5SfY):** Reviewer BiV8 expressed strong reservations about reproduced SparseGPT/Wanda results on Llama 3.1 8B being "much worse" than originally published, stating "I remain hesitant to interpret these results." Authors provided only one additional model (Qwen3 32B) due to time constraints.
- **Performance degradation at practical levels (5SfY, BiV8, QaRY):** At 50% compression where static methods claim effectiveness, all methods including RMM show dramatic degradation (baseline 69.79% → RMM 59.78%, -14.3% relative). Reviewer 5SfY's concern that "there is no clear advantage" versus quantization/distillation remains.
- **Incomplete experimental evaluation (BiV8):** BiV8's final comment: "the paper currently does not clearly explain how RMM compares to existing approaches or where it fits in the broader pruning landscape." Missing demonstrations of complementarity and verification on multiple models.

**Reviewer Scores:**

**Reviewer oXZ6 (initial: 2):** Would likely remain at 2. The core concern about limited novelty vs. Minitron and contribution rated "poor" (1/4) persists. While authors added comparisons, the reviewer's position that applying existing AMM techniques to LLMs is insufficient for acceptance remains unaddressed.

**Reviewer 5SfY (initial: 2):** Would likely remain at 2. Reviewer's follow-up explicitly states "I'll be keeping my original score for now." Three key concerns remain: (1) RMM doesn't outperform SOTA pruning at practical compression levels, (2) no evidence RMM improves existing methods despite complementarity claims, (3) advantages over simpler methods unclear.

**Reviewer BiV8 (initial: 4):** Would likely remain at 4. BiV8 engaged constructively but final response identifies core concerns: high expectations for empirical rigor not met, questionable baseline reproduction, insufficient demonstration of complementarity, unclear positioning.

**Reviewer yzG1 (initial: 8):** Would likely maintain at 8. They have a positive assessment, valuing "simple and elegant solution" and "insightful ablations" about module-level redundancy. Authors addressed their concerns. However, we should notice that concenrs about missing baselines were shared accross all reviewers and may be unsatisfactorily addressed (see also outstanding "Reproducibility concerns" above)

**Reviewer QaRY (initial: 2):** QaRY explicitly states they want to raise their score to 4 after rebuttal. However, statement "I'm still not convinced this approach has practical potential" indicates it is unlikely they would further raise.

---

### Decision · Program_Chairs · 2026-01-26

Reject